# Learning with Fredholm Kernels

**Qichao Que**   **Mikhail Belkin**   **Yusu Wang**
Department of Computer Science and Engineering
The Ohio State University
Columbus, OH 43210
{que,mbelkin,yusu}@cse.ohio-state.edu

## Abstract

In this paper we propose a framework for supervised and semi-supervised learning based on reformulating the learning problem as a regularized Fredholm integral equation. Our approach fits naturally into the kernel framework and can be interpreted as constructing new data-dependent kernels, which we call Fredholm kernels. We proceed to discuss the "noise assumption" for semi-supervised learning and provide both theoretical and experimental evidence that Fredholm kernels can effectively utilize unlabeled data under the noise assumption. We demonstrate that methods based on Fredholm learning show very competitive performance in the standard semi-supervised learning setting.

## 1   Introduction

Kernel methods and methods based on integral operators have become one of the central areas of machine learning and learning theory. These methods combine rich mathematical foundations with strong empirical performance. In this paper we propose a framework for supervised and unsupervised learning as an inverse problem based on solving the integral equation known as the Fredholm problem of the first kind. We develop regularization based algorithms for solving these systems leading to what we call Fredholm kernels.

In the basic setting of supervised learning we are given the data set $(x_i, y_i)$, where $x_i \in X, y_i \in \mathbb{R}$. We would like to construct a function $f : X \to \mathbb{R}$, such that $f(x_i) \approx y_i$ and $f$ is "nice enough" to generalize to new data points. This is typically done by choosing $f$ from a class of functions (a Reproducing Kernel Hilbert Space (RKHS) corresponding to a positive definite kernel for the kernel methods) and optimizing a certain loss function, such as the square loss or hinge loss.

In this paper we formulate a new framework for learning based on interpreting the learning problem as a Fredholm integral equation. This formulation shares some similarities with the usual kernel learning framework but unlike the standard methods also allows for easy incorporation of unlabeled data. We also show how to interpret the resulting algorithm as a standard kernel method with a non-standard data-dependent kernel (somewhat resembling the approach taken in [13]).

We discuss reasons why incorporation of unlabeled data may be desirable, concentrating in particular on what may be termed "the noise assumption" for semi-supervised learning, which is related but distint from manifold and cluster assumption popular in the semi-supervised learning literature. We provide both theoretical and empirical results showing that the Fredholm formulation allows for efficient denoising of classifiers.

To summarize, the main contributions of the paper are as follows:
(1) We formulate a new framework based on solving a regularized Fredholm equation. The framework naturally combines labeled and unlabeled data. We show how this framework can be expressed as a kernel method with a non-standard data-dependent kernel.

(2) We discuss "the noise assumption" in semi-supervised learning and provide some theoretical evidence that Fredholm kernels are able to improve performance of classifiers under this assumption. More specifically, we analyze the behavior of several versions of Fredholm kernels, based on combining linear and Gaussian kernels. We demonstrate that for some models of the noise assumption, Fredholm kernel provides better estimators than the traditional data-independent kernel and thus unlabeled data provably improves inference.

(3) We show that Fredholm kernels perform well on synthetic examples designed to illustrate the noise assumption as well as on a number of real-world datasets.

**Related work.** Kernel and integral methods in machine learning have a large and diverse literature (e.g., [12, 11]). The work most directly related to our approach is [10], where Fredholm integral equations were introduced to address the problem of density ratio estimation and covariate shift. In that work the problem of density ratio estimation was expressed as a Fredholm integral equation and solved using regularization in RKHS. This setting also relates to a line of work on on kernel mean embedding where data points are embedded in Reproducing Kernel Hilbert Spaces using integral operators with applications to density ratio estimation and other tasks [5, 6, 7]. A very interesting recent work [9] explores a shrinkage estimator for estimating means in RKHS, following the Stein-James estimator originally used for estimating the mean in an Euclidean space. The results obtained in [9] show how such estimators can reduce variance. There is some similarity between that work and our theoretical results presented in Section 4 which also show variance reduction for certain estimators of the kernel although in a different setting. Another line of related work is the class of semi-supervised learning techniques (see [15, 2] for a comprehensive overview) related to manifold regularization [1], where an additional graph Laplacian regularizer is added to take advantage of the geometric/manifold structure of the data. Our reformulation of Fredholm learning as a kernel, addressing what we called "noise assumptions", parallels data-dependent kernels for manifold regularization proposed in [13].

## 2 Fredholm Kernels

We start by formulating learning framework proposed in this paper. Suppose we are given $l$ labeled pairs $(x_1, y_1), \ldots, (x_l, y_l)$ from the data distribution $p(x, y)$ defined on $X \times Y$ and $u$ unlabeled points $x_{l+1}, \ldots, x_{l+u}$ from the marginal distribution $p_X(x)$ on $X$. For simplicity we will assume that the feature space $X$ is a Euclidean space $\mathbb{R}^D$, and the label set $Y$ is either $\{-1, 1\}$ for binary classification or the real line $\mathbb{R}$ for regression. Semi-supervised learning algorithms aim to construct a (predictor) function $f : X \to Y$ by incorporating the information of unlabeled data distribution.

To this end, we introduce the integral operator $\mathcal{K}_{p_X}$ associated with a kernel function $k(x, z)$. In our setting $k(x, z)$ does not have to be a positive semi-definite (or even symmetric) kernel.

$$\mathcal{K}_{p_X} : L^2 \to L^2 \text{ and } \mathcal{K}_{p_X} f(x) = \int k(x, z) f(z) p_X(z) dz, \tag{1}$$

where $L^2$ is the space of square-integrable functions. By the law of large numbers, the above operator can be approximated using unlabeled data from $p_X$ as

$$\mathcal{K}_{\hat{p}_X} f(x) = \frac{1}{l+u} \sum_{i=1}^{l+u} k(x, x_i) f(x_i).$$

This approximation provides a natural way of incorporating unlabeled data into algorithms. In our *Fredholm learning framework*, we will use functions in $\mathcal{K}_{p_X} \mathcal{H} = \{\mathcal{K}_{p_X} f : f \in \mathcal{H}\}$, where $\mathcal{H}$ is an appropriate Reproducing Kernel Hilbert Space (RKHS) as classification or regression functions. Note that unlike RKHS, this space of functions, $\mathcal{K}_{p_X} \mathcal{H}$, is density dependent.

In particular, this now allows us to formulate the following optimization problem for *semi-supervised* classification/regression in a way similar to many *supervised learning* algorithms:
**The Fredholm learning framework** solves the following optimization problem[1]:

$$f^* = \arg\min_{f \in \mathcal{H}} \frac{1}{l} \sum_{i=1}^{l} ((\mathcal{K}_{\hat{p}_X} f)(x_i) - y_i)^2 + \lambda \|f\|_{\mathcal{H}}^2, \tag{2}$$

The final classifier is $c(x) = (\mathcal{K}_{\hat{p}_X} f^*)(x)$, where $\mathcal{K}_{\hat{p}_X}$ is the operator defined above. Eqn 2 is a discretized and regularized version of the Fredholm integral equation $\mathcal{K}_{p_X} f = y$, thus giving the name of Fredholm learning framework.

Even though at a first glance this setting looks similar to conventional kernel methods, the extra layer introduced by $\mathcal{K}_{\hat{p}_X}$ makes significant difference, in particular, by allowing the integration of information from unlabeled data distribution. In contrast, solutions to standard kernel methods for most kernels, e.g., linear, polynomial or Gaussian kernels, are completely independent of the unlabeled data. We note that our approach is closely related to [10] where a Fredholm equation is used to estimated the density ratio for two probability distributions.

The Fredholm learning framework is a generalization of the standard kernel framework. In fact, if the kernel $k$ is the $\delta$-function, then our formulation above is equivalent to the Regularized Kernel Least Squares equation $f^* = \arg\min_{f \in \mathcal{H}} \frac{1}{l} \sum_{i=1}^{l} (f(x_i) - y_i)^2 + \lambda\|f\|_{\mathcal{H}}^2$. We could also replace the $L^2$ loss in Eqn 2 by other loss functions, such as hinge loss, resulting in a SVM-like classifier.

Finally, even though Eqn 2 is an optimization problem in a potentially infinite dimensional function space $\mathcal{H}$, a standard derivation, using the Representer Theorem (See full version for details), yields a computationally accessible solution as follows:

$$f^*(x) = \frac{1}{l+u} \sum_{j=1}^{l+u} k_{\mathcal{H}}(x, x_j) v_j, \quad \boldsymbol{v} = \left(K_{l+u}^T K_{l+u} K_{\mathcal{H}} + \lambda I\right)^{-1} K_{l+u}^T \boldsymbol{y}, \tag{3}$$

where $(K_{l+u})_{ij} = k(x_i, x_j)$ for $1 \le i \le l, 1 \le j \le l+u$, and $(K_{\mathcal{H}})_{ij} = k_{\mathcal{H}}(x_i, x_j)$ for $1 \le i, j \le l+u$. Note that $K_{l+u}$ is a $l \times (l+u)$ matrix.

**Fredholm kernels: a convenient reformulation.** In fact we will see that Fredholm learning problem induces a new data-dependent kernel, which we will refer to as *Fredholm kernel*[2]. To show this connection, we use the following identity, which can be easily verified:

$$\left(K_{l+u}^T K_{l+u} K_{\mathcal{H}} + \lambda I\right)^{-1} K_{l+u}^T = K_{l+u}^T \left(K_{l+u} K_{\mathcal{H}} K_{l+u}^T + \lambda I\right)^{-1}.$$

Define $K_F = K_{l+u} K_{\mathcal{H}} K_{l+u}^T$ to be the $l \times l$ kernel matrix associated with a new kernel defined by

$$\hat{k}_F(x, z) = \frac{1}{(l+u)^2} \sum_{i,j=1}^{l+u} k(x, x_i) k_{\mathcal{H}}(x_i, x_j) k(z, x_j), \tag{4}$$

and we consider the unlabeled data are fixed for computing this new kernel. Using this new kernel $\hat{k}_F$, the final classifying function from Eqn 3 can be rewritten as:

$$c^*(x) = \frac{1}{l+u} \sum_{i=1}^{l+u} k(x, x_i) f^*(x_i) = \sum_{s=1}^{l} \hat{k}_F(x, x_s) \alpha_s, \quad \boldsymbol{\alpha} = (K_F + \lambda I)^{-1} \boldsymbol{y}.$$

Because of Eqn 4 we will sometimes refer to the kernels $k_{\mathcal{H}}$ and $k$ as the "inner" and "outer" kernels respectively. It can be observed that this solution is equivalent to a standard kernel method, but using a new data dependent kernel $\hat{k}_F$, which we will call the *Fredholm kernel*, since it is induced from the Fredholm problem formulated in Eqn 2.

**Proposition 1.** *The Fredholm kernel defined in Eqn 4 is positive semi-definite as long as $K_{\mathcal{H}}$ is positive semi-definite for any set of data $x_1, \ldots, x_{l+u}$.*

The proof is given in the full version. The "outer" kernel $k$ does not have to be either positive definite or even symmetric. When using Gaussian kernel for $k$, discrete approximation in Eqn 4 might be unstable when the kernel width is small, so we also introduce the *normalized Fredholm kernel*,

$$\hat{k}_F^N(x, z) = \sum_{i,j=1}^{l+u} \frac{k(x, x_i)}{\sum_n k(x, x_n)} k_{\mathcal{H}}(x_i, x_j) \frac{k(z, x_j)}{\sum_n k(z, x_n)}. \tag{5}$$

It is easy to check that the resulting Fredholm kernel $\hat{k}_F^N$ is still symmetric positive semi-definite. Even though Fredholm kernel was derived using L2 loss here, it could also be derived when hinge loss is used, which will be explained in full version.

# 3    The Noise Assumption and Semi-supervised Learning

In order for unlabeled data to be useful in classification tasks it is necessary for the marginal distribution of the unlabeled data to contain information about the conditional distribution of the labels. Several ways in which such information can be encoded has been proposed including the "cluster assumption" [3] and the "manifold assumption" [1]. The cluster assumption states that a cluster (or a high density area) contains only (or mostly) points belonging to the same class. That is, if $x_1$ and $x_2$ belong to the same cluster, the corresponding labels $y_1, y_2$ should be the same. The manifold assumption assumes that the regression function is smooth with respect to the underlying manifold structure of the data, which can be interpreted as saying that the geodesic distance should be used instead of the ambient distance for optimal classification. The success of algorithms based on these ideas indicates that these assumptions do capture certain characteristics of real data. Still, better understanding of unlabeled data may still lead to progress in data analysis.

**The noise assumption.** We propose to formulate a new assumption, the "noise assumption", which is that in the neighborhood of every point, the directions with low variance (for the unlabeled data) are uninformative with respect to the class labels, and can be regarded as noise. While intuitive, as far as we know, it has not been explicitly formulated in the context of semi-supervised learning algorithms, nor applied to theoretical analysis.

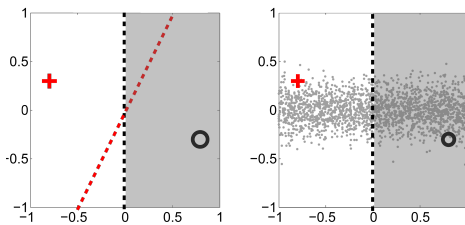

Figure 1: Left: only labelled points, and Right: with unlabelled points.

Note that even if the noise variance is small along a single direction, it could still significantly decrease the performance of a supervised learning algorithm if the noise is high-dimensional. These accumulated non-informative variations in particular increase the difficulty of learning a good classifier when the amount of labeled data is small. The first figure on right illustrates the issue of noise with two labeled points. The seemingly optimal classification boundary (the red line) differs from the correct one (in black) due to the noisy variation along the $y$ axis for the two labeled points. Intuitively unlabeled data shown in the right panel of Figure 1 can be helpful in this setting as low variance directions can be estimated locally such that algorithms could suppress the influences of the noisy variation when learning a classifier.

**Connection to cluster and manifold assumptions.** The noise assumption is compatible with the manifold assumption within the manifold+noise model. Specifically, we can assume that the functions of interest vary along the manifold and are constant in the orthogonal direction. Alternatively, we can think of directions with high variance as "signal/manifold" and directions with low variance as "noise". We note that the noise assumption does not require the data to conform to a low-dimensional manifold in the strict mathematical sense of the word. The noise assumption is orthogonal to the cluster assumption. For example, Figure 1 illustrates a situation where data has no clusters but the noise assumption applies.

# 4    Theoretical Results for Fredholm Kernels

Non-informative variation in data could degrade traditional supervised learning algorithms. We will now show that Fredholm kernels can be used to replace traditional kernels to inject them with "noise-suppression" power with the help of unlabeled data. In this section we will present two views to illustrate how such noise suppression can be achieved. Specifically, in Section 4.1) we show that under certain setup, linear Fredholm kernel suppresses principal components with small variance. In Section 4.2) we prove that under certain conditions we are able to provide good approximations to the "true" kernel on the hidden underlying space.

To make our arguments more clear, we assume that there are infinite amount of unlabelled data; that is, we know the marginal distribution of data exactly. We will then consider the following continuous versions of the un-normalized and normalized Fredholm kernels as in Eqn 4 and 5:

$$k_F^U(x, z) = \int \int k(x, u) k_{\mathcal{H}}(u, v) k(z, v) p(u) p(v) du dv \tag{6}$$

$$k_F^N(x, z) = \int \int \frac{k(x, u)}{\int k(x, w) p(w) dw} k_{\mathcal{H}}(u, v) \frac{k(z, v)}{\int k(z, w) p(w) dw} p(u) p(v) du dv. \tag{7}$$

Note, in the above equations and in what follows, we sometimes write $p$ instead of $p_X$ for the marginal distribution when its choice is clear from context. We will typically use $k_F$ to denote appropriate normalized or unnormalized kernels depending on the context.

## 4.1 Linear Fredholm kernels and inner products

For this section, we consider the unnormalized Fredholm kernel, that is $k_F = k_F^U$. If the "outer" kernel $k(u, v)$ is linear, i.e. $k(u, v) = \langle u, v \rangle$, the resulting Fredholm kernel can be viewed as an inner product. Specifically, the un-normalized Fredholm kernel from Eqn 6 can be rewritten as:

$$k_F(x, z) = x^T \Sigma_F z, \quad \text{where} \quad \Sigma_F = \int \int u k_{\mathcal{H}}(u, v) v^T p(u) p(v) du dv.$$

Thus $k_F(x, z)$ is simply an inner product which depends on both the unlabeled data distribution $p(x)$ and the "inner" kernel $k_{\mathcal{H}}$. This inner product re-weights the standard norm in feature space based on variances along the principal directions of the matrix $\Sigma_F$. We show that for the model when unlabeled data is sampled from a normal distribution this kernel can be viewed as a "soft thresholding" PCA, suppressing the directions with low variance. Specifically, we have the following[3]

**Theorem 2.** *Let* $k_{\mathcal{H}}(x, z) = \exp\left(-\frac{\|x-z\|^2}{2t}\right)$ *and assume the distribution* $p_X$ *for unlabeled data is a single multi-variate normal distribution,* $N(\mu, diag(\sigma_1^2, \ldots, \sigma_d^2))$. *We have*

$$\Sigma_F = \left(\prod_{d=1}^{D} \sqrt{\frac{t}{2\sigma_d^2 + t}}\right)\left(\mu\mu^T + diag\left(\frac{\sigma_1^4}{2\sigma_1^2 + t}, \ldots, \frac{\sigma_D^4}{2\sigma_D^2 + t}\right)\right).$$

Assuming that the data is mean-subtracted, i.e. $\mu = 0$, we see that $x^T \Sigma_F z$ re-scales the projections along the principal components when computing the inner product; that is, the rescaling factor for the $i$-th principal direction is $\sqrt{\frac{\sigma_i^4}{2\sigma_i^2 + t}}$.

Note that this rescaling factor $\frac{\sigma_i^4}{2\sigma_i^2 + t} \approx 0$ when $\sigma_i^2 \ll t$. On the other hand when $\sigma_i^2 \gg t$, we have that $\frac{\sigma_i^4}{2\sigma_i^2 + t} \approx \frac{\sigma_i^2}{2}$. Hence $t$ can be considered as a *soft threshold* that eliminates the effects of principal components with small variances. When $t$ is small the rescaling factors are approximately proportional to $diag(\sigma_1^2, \sigma_2^2, \ldots, \sigma_D^2)$, in which case $\Sigma_F$ is is proportional to the covariance matrix of the data $XX^T$.

## 4.2 Kernel Approximation With Noise

We have seen that one special case of Fredholm kernel could achieve the effect of principal components re-scaling by using linear kernel as the "outer" kernel $k$. In this section we give a more general interpretation of noise suppression by the Fredholm kernel.

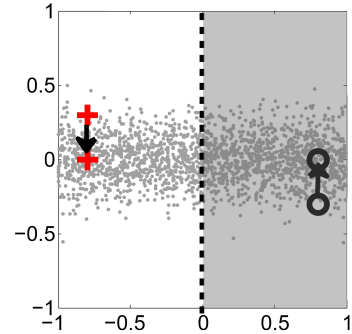

First, we give a simple senario to provide some intuition behind the definition of Fredholm kernle. Consider a standard supervised learning setting which uses the solution $f^* = \arg\min_{f \in \mathcal{H}} \frac{1}{l}\sum_{i=1}^{l}(f(x_i) - y_i)^2 + \lambda\|f\|_{\mathcal{H}}^2$ as the classifier. Let $k_{\mathcal{H}}^{\text{target}}$ denote the ideal kernel that we intend to use on the clean data, which we call the *target kernel* from now on. Now suppose what we have are two noisy labelled points $x_e$ and $z_e$ for "true" data $\bar{x}$ and $\bar{z}$, i.e. $x_e = \bar{x} + \varepsilon_x$, $z_e = \bar{z} + \varepsilon_z$. The evaluation of $k_{\mathcal{H}}^{\text{target}}(x_e, z_e)$ can be quite different from the true signal $k_{\mathcal{H}}^{\text{target}}(\bar{x}, \bar{z})$, leading to an suboptimal final classifier (the red line in Figure 1 (a)). On the other hand, now consider the Fredholm kernel from Eqn 6 (or similarly from Eqn 7): $k_F(x_e, z_e) = \int \int k(x_e, u)p(u) \cdot k_{\mathcal{H}}(u, v) \cdot k(z_e, v)p(v)dudv$, and set the outer kernel $k$ to be the Gaussian kernel, and the inner kernel $k_{\mathcal{H}}$ to be the same as target kernel $k_{\mathcal{H}}^{\text{target}}$. We can think of $k_F(x_e, z_e)$ as an averaging of $k_{\mathcal{H}}(u, v)$ over all possible pairs of data $u, v$, weighted by $k(x_e, u)p(u)$ and $k(z_e, v)p(v)$ respectively. Specifically, points

that are close to $x_e$ (resp. $z_e$) with high density will receive larger weights. Hence the weighted averages will be biased towards $\bar{x}$ and $\bar{z}$ respectively (which presumably lie in high density regions around $x_e$ and $z_e$). The value of $k_F(x_e, z_e)$ tends to provide a more accurate estimate of $k_{\mathcal{H}}(\bar{x}, \bar{z})$. See the right figure for an illustration where the arrows indicate points with stronger influences in the computation of $k_F(x_e, z_e)$ than $k_{\mathcal{H}}(x_e, z_e)$. As a result, the classifier obtained using the Fredholm kernel will also be more resilient to noise and closer to the optimum.

The Fredholm learning framework is rather flexible in terms of the choices of kernels $k$ and $k_{\mathcal{H}}$. In the remainder of this section, we will consider a few specific scenarios and provide quantitative analysis to show the noise robustness of the Fredholm kernel.

**Problem setup.** Assume that we have a ground-truth distribution over the subspace spanned by the first $d$ dimension of the Euclidean space $\mathbb{R}^D$. We will assume that this distribution is a single Gaussian $N(0, \lambda^2 I_d)$. Suppose this distribution is corrupted with Gaussian noise along the orthogonal subspace of dimension $D - d$. That is, for any "true" point $\bar{x}$ drawn from $N(0, \lambda^2 I_d)$, its observation $x_e$ is drawn from $N(\bar{x}, \sigma^2 I_{D-d})$. Since the noise lies in a space orthogonal to data distribution, this means that any observed point, labelled or unlabeled, is sampled from $p_X = N(0, diag(\lambda^2 I_d, \sigma^2 I_{D-d}))$. We will show that Fredholm kernel provides a better approximation to the "original" kernel given unlabeled data than simply computing the kernel of noisy points. We choose this basic setting to be able to state the theoretical results in a clean manner. Even though this is a Gaussian distribution over a linear subspace with noise, this framework has more general implications since local neighborhoods of manifolds are (almost) linear spaces.

**Note:** In this section we use normalized Fredholm kernel given in Eqn 7, that is $k_F = k_F^N$ for now on. Un-normalized Fredholm kernel displays similar behavior, while the bounds are trickier.

**Linear Kernel.** First we consider the case where the target kernel $k_{\mathcal{H}}^{\text{target}}(u, v)$ is the linear kernel, $k_{\mathcal{H}}^{\text{target}}(u, v) = u^T v$. We will set $k_{\mathcal{H}}$ in Fredholm kernel to also be linear, and $k$ to be the Gaussian kernel $k(u, v) = e^{-\frac{\|u-v\|^2}{2t}}$ We will compare $k_F(x_e, z_e)$ with the target kernel on the two observed points, that is, with $k_{\mathcal{H}}^{\text{target}}(x_e, z_e)$. The goal is to estimate $k_{\mathcal{H}}^{\text{target}}(\bar{x}, \bar{z})$. We will see that (1) both $k_F(x_e, z_e)$ and (appropriately scaled) $k_{\mathcal{H}}(x_e, z_e)$ are unbiased estimators of $k_{\mathcal{H}}^{\text{target}}(\bar{x}, \bar{z})$, however (2) the variance of $k_F(x_e, z_e)$ is smaller than that of $k_{\mathcal{H}}^{\text{target}}(x_e, z_e)$, making it a more precise estimator.

**Theorem 3.** *Suppose the probability distribution for the unlabeled data* $p_X = N(\mathbf{0}, diag(\lambda^2 I_d, \sigma^2 I_{D-d}))$. *For Fredholm kernel defined in Eqn 7, we have*

$$\mathbb{E}_{x_e, z_e}(k_{\mathcal{H}}^{target}(x_e, z_e)) = \mathbb{E}_{x_e, z_e}\left(\left(\frac{t + \lambda^2}{\lambda^2}\right)^2 k_F(x_e, z_e)\right) = \bar{x}^T \bar{z}$$

*Moreover, when* $\lambda > \sigma$, $Var_{x_e, z_e}\left(\left(\frac{t+\lambda^2}{\lambda^2}\right)^2 k_F(x_e, z_e)\right) < Var_{x_e, z_e}(k_{\mathcal{H}}^{target}(x_e, z_e))$.

**Remark:** Note that we have a normalization constant for the Fredholm kernel to make it an unbiased estimator of $\bar{x}^T \bar{z}$. In practice, choosing normalization is subsumed in selecting the regularization parameter for kernel methods.

Thus we can see the Fredholm kernel provides an approximation of the "true" linear kernel, but with smaller variance compared to the actual linear kernel on noisy data.

**Gaussian Kernel.** We now consider the case where the target kernel is the Gaussian kernel: $k_{\mathcal{H}}^{\text{target}}(u, v) = \exp\left(-\frac{\|u-v\|^2}{2r}\right)$. To approximate this kernel, we will set both $k$ and $k_{\mathcal{H}}$ to be Gaussian kernels. To simplify the presentation of results, we assume that $k$ and $k_{\mathcal{H}}$ have the same kernel width $t$. The resulting Fredholm kernel turns out to also be a Gaussian kernel, whose kernel width depends on the choice of $t$.

Our main result is the following. Again, similar to the case of linear kernel, the Fredholm estimation $k_F(x_e, z_e)$ and $k_{\mathcal{H}}^{\text{target}}(x_e, z_e)$ are both unbiased estimator for the target $k_{\mathcal{H}}^{\text{target}}(\bar{x}, \bar{z})$ up to a constant; but $k_F(x_e, z_e)$ has a smaller variance.

**Theorem 4.** *Suppose the probability distribution for the unlabeled data* $p_X = N(\mathbf{0}, diag(\lambda^2 I_d, \sigma^2 I_{D-d}))$. *Given the target kernel* $k_{\mathcal{H}}^{target}(u, v) = \exp\left(-\frac{\|u-v\|^2}{2r}\right)$ *with kernel width* $r > 0$, *we can choose* $t$, *given by the equation* $\frac{t(t+\lambda^2)(t+3\lambda^2)}{\lambda^4} = r$, *and two scaling*

*constants $c_1, c_2$, such that*

$$\mathbb{E}_{x_e, z_e}(c_1^{-1} k_{\mathcal{H}}^{target}(x_e, z_e)) = \mathbb{E}_{x_e, z_e}(c_2^{-1} k_F(x_e, z_e)) = k_{\mathcal{H}}^{target}(\bar{x}, \bar{z}).$$

*and when $\lambda > \sigma$, we have $Var_{x_e, z_e}(c_1^{-1} k_{\mathcal{H}}^{target}(x_e, z_e)) > Var_{x_e, z_e}(c_2^{-1} k_F(x_e, z_e))$.*

**Remark.** In practice, when applying kernel methods for real world applications, optimal kernel width $r$ is usually unknown and chosen by cross-validation or other methods. Similarly, for our Fredholm kernel, one can also use cross-validation to choose the optimal $t$ for $k_F$.

## 5   Experiments

Using linear and Gaussian kernel for $k$ or $k_{\mathcal{H}}$ respectively, we will define three instances of the Fredholm kernel as follows.

(1) FredLin1: $k(x, z) = x^T z$ and $k_{\mathcal{H}}(x, z) = \exp\left(-\frac{\|x-z\|^2}{2r}\right)$.

(2) FredLin2: $k(x, z) = \exp\left(-\frac{\|x-z\|^2}{2r}\right)$ and $k_{\mathcal{H}}(x, z) = x^T z$.

(3) FredGauss: $k(x, z) = k_{\mathcal{H}}(x, z) = \exp\left(-\frac{\|x-z\|^2}{2r}\right)$.

For the kernels in (2) and (3) that use the Gaussian kernel as outside kernel $k$ we can also define their normalized version, which we will denote by by FredLin2(N) and FredGauss(N) respectively.

**Synthetic examples. Noise and cluster assumptions.**

To isolate the ability of Fredholm kernels to deal with noise from the cluster assumption, we construct two synthetic examples that violate the cluster assumption, shown in Figure 2. The figures show first two dimensions, with multi-variate Gaussian noise with variance $\sigma^2 = 0.01$ in $\mathbb{R}^{100}$ added. The classification boundaries are indicated by the color. For each class, we provide several labeled points and large amount of unlabeled data. Note that the classification boundary in the "circle" example is non-linear.

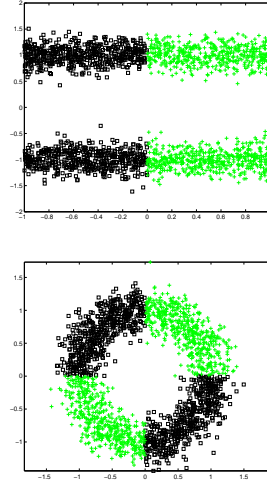

We compare Fredholm kernel based classifier with RLSC (Regularized Least Squares Classifier), and two widely used semi-supervised methods, the transductive support vector machine and LapRLSC. Since the examples violate the cluster assumption, the two existing semi-supervised learning algorithms, Transductive SVM and LapRLSC, should not gain much from the unlabeled data. For TSVM, we use the primal TSVM proposed in [4], and we will use the implementation of LapRLSC given in [1]. Different numbers of labeled points are given for each class, together with another

Figure 2: Noise but not cluster assumption. Gaussian noise in $\mathbb{R}^{100}$ is added. Linear (above) and non-linear (below) class boundaries.

2000 unlabeled points. To choose the optimal parameters for each method, we pick the parameters based on their performance on the validation set, while the final classification error is computed on the held-out testing data set. Results are reported in Table 1 and 2, in which Fredholm kernels show clear improvement over other methods for synthetic examples in term of classification error.

**Real-world Data Sets.** Unlike artificial examples, it is usually difficult to verify whether certain assumptions are satisfied in real-world problems. In this section, we examine the performance of Fredholm kernels on several real-world data sets and compare it with the baseline algorithms mentioned above.

**Linear Kernels.** Here we consider text categorization and sentiment analysis, where linear methods are known to perform well. We use the following data (represented by TF-IDF features):
(1) 20 news group: it has 11269 documents with 20 classes, and we select the first 10 categories for our experiment. (2) Webkb: the original data set contains 7746 documents with 7 unbalanced classes, and we pick the two largest classes with 1511 and 1079 instances respectively. (3) IMDB movie review: it has 1000 positive reviews and 1000 negative reviews of movie on IMDB.com. (4) Twitter sentiment data from Sem-Eval 2013: it contains 5173 tweets, with positive, neural and negative sentiment. We combine neutral and negative classes to set up a binary classification problem. Results are reported in Table 3. In Table4, we use WebKB as an example to illustrate the change of the performance as number of labeled points increases.

| Number | Methods(Linear) | | | | |
|---|---|---|---|---|---|
| of Labeled | RLSC | TSVM | LapRLSC | FredLin1 | FredLin2(N) |
| 8 | 10.0($\pm$ 3.9) | 5.2($\pm$ 2.2) | 10.0($\pm$ 3.5) | **3.7**($\pm$ 2.6) | 4.5($\pm$ 2.1) |
| 16 | 9.1($\pm$ 1.9) | 5.1($\pm$ 1.1) | 9.1($\pm$ 2.2) | **2.9**($\pm$ 2.0) | 3.6($\pm$ 1.9) |
| 32 | 5.8($\pm$ 3.2) | 4.5($\pm$ 0.8) | 6.0($\pm$ 3.2) | **2.3**($\pm$ 2.3) | 2.6($\pm$ 2.2) |

Table 1: Prediction error of different classifiers for the"two lines" example.

| Number | Methods(Gaussian) | | | |
|---|---|---|---|---|
| of Labeled | K-RLSC | TSVM | LapRLSC | FredGauss(N) |
| 16 | 17.4($\pm$ 5.0) | 32.2($\pm$ 5.2) | 17.0($\pm$ 4.6) | **7.1**($\pm$ 2.4) |
| 32 | 16.5($\pm$ 7.1) | 29.9($\pm$ 9.3) | 18.0($\pm$ 6.8) | **6.0**($\pm$ 1.6) |
| 64 | 8.7($\pm$ 1.7) | 20.3($\pm$ 4.2) | 9.7($\pm$ 2.0) | **5.5**($\pm$ 0.7) |

Table 2: Prediction error of different classifiers for the "circle" example.

**Gaussian Kernel.** We test our methods on hand-written digit recognition. The experiment use subsets of two handwriting digits data sets MNIST and USPS: (1) the one from MNIST contains 10k digits in total with balanced examples for each class, and the one for USPS is the original testing set containing about 2k images. The pixel values are normalized to $[0, 1]$ as features. Results are reported in Table 5. In Table 6, we show that as we add additional Gaussian noise to MNIST data, Fredholm kernels start to show significant improvement.

| Data Set | Methods(Linear) | | | | |
|---|---|---|---|---|---|
| | RLSC | TSVM | FredLin1 | FredLin2 | FredLin2(N) |
| Webkb | 16.9($\pm$ 1.4) | 12.7($\pm$ 0.8) | 13.0($\pm$ 1.3) | **12.0**($\pm$ 1.6) | **12.0**($\pm$ 1.6) |
| 20news | 22.2($\pm$ 1.0) | 21.0($\pm$ 0.9) | **20.5** ($\pm$ 0.7) | **20.5** ($\pm$0.7) | **20.5**($\pm$ 0.7) |
| IMDB | 30.0($\pm$ 2.0) | 20.2($\pm$ 2.6) | **19.9**($\pm$ 2.3) | 21.7($\pm$ 2.9) | 21.7($\pm$ 2.7) |
| Twitter | 38.7($\pm$ 1.1) | 37.6($\pm$ 1.4) | **37.4**($\pm$ 1.2) | **37.4**($\pm$ 1.2) | 37.5($\pm$ 1.2) |

Table 3: The error of various methods on the text data sets. 20 labeled data per class are given with rest of the data set as unlabeled points. Optimal parameter for each method are used.

| Number | Methods(Linear) | | | | |
|---|---|---|---|---|---|
| of Labeled | RLSC | TSVM | FredLin1 | FredLin2 | FredLin2(N) |
| 10 | 20.7($\pm$ 2.4) | **13.5**($\pm$ 0.5) | 14.8($\pm$ 2.4) | 14.6($\pm$ 2.4) | 14.6($\pm$ 2.3) |
| 20 | 16.9($\pm$ 1.4) | 12.7($\pm$ 0.8) | 13.0($\pm$ 1.3) | **12.0**($\pm$ 1.6) | **12.0**($\pm$ 1.6) |
| 80 | 10.9($\pm$ 1.4) | 9.7($\pm$ 1.0) | 8.1($\pm$ 1.0) | **7.9**($\pm$ 0.9) | **7.9**($\pm$ 0.9) |

Table 4: Prediction error on Webkb with different number of labeled points.

| Data Set | Methods(Gaussian) | | | |
|---|---|---|---|---|
| | K-RLSC | LapRLSC | FredGauss | FredGauss(N) |
| USPST | 11.8($\pm$ 1.4) | **10.2** ($\pm$0.5) | 12.4($\pm$ 1.8) | 10.8($\pm$ 1.1) |
| MNIST | 14.3($\pm$ 1.2) | **8.6**($\pm$ 1.2) | 12.2($\pm$1.0) | 13.0($\pm$ 0.9) |

Table 5: Prediction error of nonlinear classifiers on the MNIST and USPS. 20 labeled data per class are given with rest of the data set as unlabeled points. Optimal parameter for each method are used.

| Number | Methods(Gaussian) | | | |
|---|---|---|---|---|
| of Labeled | K-RLSC | LapRLSC | FredGauss | FredGauss(N) |
| 10 | 34.1($\pm$ 2.1) | 35.6 ($\pm$3.5) | **27.9**($\pm$ 1.6) | 29.0($\pm$ 1.5) |
| 20 | 27.2($\pm$ 1.1) | 27.3 ($\pm$1.8) | **21.9**($\pm$ 1.2) | 22.9($\pm$ 1.2) |
| 40 | 20.0($\pm$ 0.7) | 20.3 ($\pm$0.8) | **17.3**($\pm$ 0.5) | 18.4($\pm$ 0.4) |
| 80 | 15.6($\pm$ 0.4) | 15.6 ($\pm$0.5) | **14.8**($\pm$ 0.6) | 15.4($\pm$ 0.5) |

Table 6: The prediction error of nonlinear classifiers on MNIST corrupted with Gaussian noise with standard deviation 0.3, with different numbers of labeled points, from 10 to 80. Optimal parameter for each method are used.

**Acknowledgments.** The work was partially supported by NSF Grants CCF-1319406 and RI 1117707. We thank the anonymous NIPS reviewers for insightful comments.

## Footnotes

[1]We will be using the square loss to simplify the exposition. Other loss functions can also be used in Eqn 2.

[2] We note that the term Fredholm Kernel has been used in mathematics ([8], page 103) and also in a different learning context [14]. Our usage represents a different object.

[3]The proof of this and other results can be found in the full version.

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
