[Supplementary Material]

# Learning with Fredholm Kernels

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

where $L^2$ is the space of square-integrable functions. As usual, by the law of large number, the above operator can be approximated by the unlabeled data from $p_X$ as follows,

$$\mathcal{K}_{\hat{p}_X} f(x) = \frac{1}{l+u} \sum_{i=1}^{l+u} k(x, x_i) f(x_i). \tag{2}$$

This approximation provides a natural way of incorporating unlabeled data into algorithms. In our *Fredholm learning framework*, we will use functions in $\mathcal{K}_{p_X} \mathcal{H} = \{\mathcal{K}_{p_X} f : f \in \mathcal{H}\}$, where $\mathcal{H}$ is an appropriate Reproducing Kernel Hilbert Space (RKHS) as classification or regression functions. Note that unlike RKHS, this space of functions, $\mathcal{K}_{p_X} \mathcal{H}$, is density dependent.

In particular, this now allows us to formulate the following optimization problem for *semi-supervised* classification/regression in a way similar to many *supervised learning* algorithms:

**The Fredholm learning framework** solves the following optimization problem[1]:

$$f^* = \arg\min_{f \in \mathcal{H}} \frac{1}{l} \sum_{i=1}^{l} ((\mathcal{K}_{\hat{p}_X} f)(x_i) - y_i)^2 + \lambda \|f\|_{\mathcal{H}}^2, \tag{3}$$

The final classifier is $c(x) = (\mathcal{K}_{\hat{p}_X} f^*)(x)$, where $\mathcal{K}_{\hat{p}_X}$ is the operator defined above. Eqn 3 is a discretized and regularized version of the Fredholm integral equation $\mathcal{K}_{p_X} f = y$, thus giving the name of Fredholm learning framework.

Even though at first glance this setting looks similar to conventional kernel methods, the extra layer introduced by $\mathcal{K}_{\hat{p}_X}$ makes significant difference, in particular, by allowing the integration of information from unlabelled data distribution. In contrast, solutions to kernel method for most kernels, e.g., linear, polynomial or Gaussian kernels, are completely independent of the unlabeled data. We note that our approach is closely related to [10] where a Fredholm equation is used to estimated the density ratio for two probability distributions.

Our Fredholm learning framework is a generalization of the standard kernel framework. In fact, if the kernel $k$ is the $\delta$-function, then our formulation above is equivalent to the standard Regularized Least Squares equation $f^* = \arg\min_{f \in \mathcal{H}} \frac{1}{l} \sum_{i=1}^{l} (f(x_i) - y_i)^2 + \lambda \|f\|_{\mathcal{H}}^2$. We could also replace the $L^2$ loss in Eqn 3 by other loss functions, such as hinge loss, resulting in a SVM-like classifier.

Finally, even though Eqn 3 is an optimization problem in a potentially infinite dimensional function space $\mathcal{H}$, we have the following lemma that allows us to apply the Representer Theorem to get a computationally accessible solution.

**Lemma 1.** *Given the definition of $\mathcal{K}_{\hat{p}_X}$ in Eqn 2, the solution to Eqn 3 is of the form,*

$$f^*(x) = \frac{1}{l+u} \sum_{j=1}^{l+u} k_{\mathcal{H}}(x, x_j) v_j,$$

*for some $\boldsymbol{v} \in \mathbb{R}^{l+u}$.*

As the proof of the above lemma is similar to that of the standard representer theorem, we put the proof in the appendix. Using the above Representer Theorem, we could transform Eqn 3 into quadratic optimization in a finite dimensional space. We can get have a closed form solution for Eqn 3 as follows:

$$f^*(x) = \frac{1}{l+u} \sum_{j=1}^{l+u} k_{\mathcal{H}}(x, x_j) v_j, \quad \boldsymbol{v} = \left(K_{l+u}^T K_{l+u} K_{\mathcal{H}} + \lambda I\right)^{-1} K_{l+u}^T \boldsymbol{y}, \tag{4}$$

where $(K_{l+u})_{ij} = k(x_i, x_j)$ for $1 \le i \le l, 1 \le j \le l+u$, and $(K_{\mathcal{H}})_{ij} = k_{\mathcal{H}}(x_i, x_j)$ for $1 \le i, j \le l+u$. Note that $K_{l+u}$ is a $l \times (l+u)$ matrix.

**Fredholm kernels: a convenient reformulation.** Interestingly, this Fredholm learning problem actually induces a new data-dependent kernel, which we will refer to as *Fredholm kernel*[2]. To show this connection, first observe the following identity, which can be easily verified:

**Claim 2.** *Matrix Inversion Identity*

$$\left(K_{l+u}^T K_{l+u} K_{\mathcal{H}} + \lambda I\right)^{-1} K_{l+u}^T = K_{l+u}^T \left(K_{l+u} K_{\mathcal{H}} K_{l+u}^T + \lambda I\right)^{-1}.$$

Define $K_F = K_{l+u} K_{\mathcal{H}} K_{l+u}^T$ to be the $l \times l$ kernel matrix associated with a new kernel defined by

$$\hat{k}_F(x, z) = \frac{1}{(l+u)^2} \sum_{i,j=1}^{l+u} k(x, x_i) k_{\mathcal{H}}(x_i, x_j) k(z, x_j), \tag{5}$$

and we consider the unlabeled data are fixed for computing this new kernel. Using this new kernel $\hat{k}_F$, the final classifying function $c^*$ defined using the solution given in Eqn 4 can be rewritten as:

$$c^*(x) = \frac{1}{l+u} \sum_{i=1}^{l+u} k(x, x_i) f^*(x_i) = \sum_{s=1}^{l} \hat{k}_F(x, x_s) \alpha_s, \quad \boldsymbol{\alpha} = (K_F + \lambda I)^{-1} \boldsymbol{y}.$$

Because of Eqn 5 we will sometimes refer to the kernels $k_{\mathcal{H}}$ and $k$ as the "inner" and "outer" kernels respectively.

It can be observed that this learning algorithm can be considered as a case of the standard kernel method, but using a new data dependent kernel $\hat{k}_F$, which we will call the *Fredholm kernel*, since it is induced from the Fredholm problem formulated in Eqn 3. And the following proposition shows that this definition gives a positive semi-definite kernel.

**Proposition 3.** *The Fredholm kernel defined in Eqn 5 is positive semi-definite if $k_{\mathcal{H}}$ is a positive semi-definite kernel.*

The proof is given in the appendix. The "outer" kernel $k$ does not have to be either positive definite or even symmetric. When using Gaussian kernel for $k$, discrete approximation in Eqn 5 might be unstable when the kernel width is small, so we also introduce the *normalized Fredholm kernel*,

$$\hat{k}_F^N(x,z) = \frac{1}{(l+u)^2} \sum_{i,j=1}^{l+u} \frac{k(x,x_i)}{\sum_n k(x,x_n)} k_{\mathcal{H}}(x_i,x_j) \frac{k(z,x_j)}{\sum_n k(z,x_n)}. \tag{6}$$

It is easy to check that the resulting Fredholm kernel $\hat{k}_F^N$ is still symmetric and positive semi-definite.

**Using Hinge Loss** Other than L2 loss we use above, hinge loss can also be used for our Fredholm learning framework. In this section, we explain how Fredholm kernel could be derived when using hinge loss. Plugging the hinge loss into Eqn 3, we have

$$f^* = \arg\min_{f \in \mathcal{H}} \frac{1}{l} \sum_{i=1}^{l} \max(0, 1 - y_i \cdot (\mathcal{K}_{\hat{p}_X} f)(x_i)) + \lambda \|f\|_{\mathcal{H}}^2. \tag{7}$$

Like the Representer Theorem, we proved in Lemma 1, the solution function $f$ is always of the form

$$f(x) = \sum_{i=1}^{l+u} v_i k_{\mathcal{H}}(x,x_i).$$

Thus, $\|f\|_{\mathcal{H}}^2 = \boldsymbol{v}^T K_{\mathcal{H}} \boldsymbol{v}$, where $K_{\mathcal{H}}$ is the kernel matrix.

And we only consider the evaluation of $f$ at the data points, let $\boldsymbol{f} = [f(x_1), \ldots, f(x_{l+u})] = K_{\mathcal{H}} \boldsymbol{v}$. Now we can vectorize $(\mathcal{K}_{\hat{p}_X} f)(x_i)$ as well, by letting $\boldsymbol{k}_i = [\frac{1}{l+u}k(x_i,x_1), \ldots, \frac{1}{l+u}k(x_i,x_{l+u})]$. Thus $\mathcal{K}_{\hat{p}_X} f(x_i) = \frac{1}{l+u} \sum_{j=1}^{l+u} k(x_i,x_j) f(x_j) = \boldsymbol{k}_i^T \boldsymbol{f} = \boldsymbol{k}_i^T K_{\mathcal{H}} \boldsymbol{v}$.

And the optimization problem using hinge loss in Eqn 7 is equivalent to the following problem with slack variables $\xi_i$,

$$\min_{f \in \mathcal{H}} \frac{1}{2} \boldsymbol{v}^T K_{\mathcal{H}} \boldsymbol{v} + C \sum_i \xi_i$$

$$\text{s.t.} \quad y_i \cdot (\boldsymbol{k}_i^T K_{\mathcal{H}} \boldsymbol{v}) \geq 1 - \xi_i$$
$$\xi_i \geq 0 \quad \text{for } i = 1, \ldots, l$$

To solve the above problem, we introduce the Lagrangian multiplier,

$$\mathcal{L}(\boldsymbol{v}, \xi, \alpha, \gamma) = \frac{1}{2} \boldsymbol{v}^T K_{\mathcal{H}} \boldsymbol{v} + C \sum_i \xi_i - \sum_i \alpha_i (y_i \cdot (\boldsymbol{k}_i K_{\mathcal{H}} \boldsymbol{v}) - 1 + \xi_i) - \sum_i \gamma_i \xi_i$$

By the KKT condition in the theory of convex optimization, we have

$$\boldsymbol{v} = \sum_i \alpha_i y_i \boldsymbol{k}_i, \quad \alpha_i = C - \gamma_i$$

Using this, we have the dual problem of the original problem in Eqn 7,

$$\max_{\alpha} \sum_i \alpha_i - \frac{1}{2} \sum_{i,j} \alpha_i \alpha_j y_i y_j \boldsymbol{k}_i^T K_{\mathcal{H}} \boldsymbol{k}_j$$

$$s.t. \quad 0 \leq \alpha_i \leq C.$$

It is equivalent to using Fredholm kernel for regular support vector machine, because $\boldsymbol{k}_i^T K_{\mathcal{H}} \boldsymbol{k}_j = k_F(x_i,x_j)$ according to the definition of Fredholm kernel in Eqn 5.

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

**Problem setup.** Assume that we have a ground-truth distribution over the subspace spanned by the first $d$ dimension of the Euclidean space $\mathbb{R}^D$. We will assume that this ground-truth distribution is a single Gaussian $N(0, \lambda^2 I_d)$. Now imagine that this ground-truth distribution is corrupted with Gaussian noise along the orthogonal subspace of dimension $D - d$. That is, for any observed point $x_e$, it could be decomposed into $\bar{x} + \varepsilon_x$, where the signal $\bar{x}$ is drawn from $N(0, \lambda^2 I_d)$, and the noise $\varepsilon_x$ is drawn from $N(\mathbf{0}, \sigma^2 I_{D-d})$ over the orthogonal space. Thus any observed point, labelled or unlabelled, is sampled from $p_X = N(0, diag(\lambda^2 I_d, \sigma^2 I_{D-d}))$, with the first $d$ dimensions as signals and the rest corrupted by noises.

We will show that Fredholm kernel provides a better approximation to the "original" kernel given both labeled and unlabeled data than directly computing the kernel evaluation at noisy labeled points.

We choose this simple setting so as to be able to state the theoretical results in a clean manner. Even though this is just a Gaussian distribution over a linear subspace with noise this framework can be generalized since local neighborhoods of a Riemannian manifold can be approximated by linear spaces.

**Note.** In this section, we use the normalized Fredholm kernel given in Eqn 9 for simplicity, that is $k_F = k_F^N$ for now on. Un-normalized Fredholm kernel displays similar behavior, however, the theoretical bounds are more complicated.

### 4.2.1   Linear Kernel

First we consider the case where the target kernel $k_{\mathcal{H}}^{\text{target}}(u, v)$ is the linear kernel, $k_{\mathcal{H}}^{\text{target}}(u, v) = u^T v$. We will set $k_{\mathcal{H}}$ in Fredholm kernel to also be linear, and $k$ to be the Gaussian kernel $k(u, v) = e^{-\frac{\|u-v\|^2}{2t}}$ We will compare $k_F(x_e, z_e)$ with the target kernel on the two observed points, that is, with $k_{\mathcal{H}}^{\text{target}}(x_e, z_e)$. The goal is to estimate $k_{\mathcal{H}}^{\text{target}}(\bar{x}, \bar{z})$. We will see that (1) both $k_F(x_e, z_e)$ and (appropriately scaled) $k_{\mathcal{H}}(x_e, z_e)$ are unbiased estimators of $k_{\mathcal{H}}^{\text{target}}(\bar{x}, \bar{z})$, however (2) the variance of $k_F(x_e, z_e)$ is smaller than that of $k_{\mathcal{H}}^{\text{target}}(x_e, z_e)$, making it a more precise estimator.

**Theorem 5.** *Suppose the probability distribution for the data is $p_X = N(\mathbf{0}, diag(\lambda^2 I_d, \sigma^2 I_{D-d}))$. For Fredholm kernel defined in Eqn 9, we have*

$$\mathbb{E}_{x_e, z_e}(k_{\mathcal{H}}^{target}(x_e, z_e)) = \mathbb{E}_{x_e, z_e}\left(\left(\frac{t + \lambda^2}{\lambda^2}\right)^2 k_F(x_e, z_e)\right) = \bar{x}^T \bar{z}$$

*Moreover, when $\lambda > \sigma$, $Var_{x_e, z_e}\left(\left(\frac{t+\lambda^2}{\lambda^2}\right)^2 k_F(x_e, z_e)\right) < Var_{x_e, z_e}(k_{\mathcal{H}}^{target}(x_e, z_e))$.*

**Remark:** Note that we have a normalization constant for the Fredholm kernel to make it an unbiased estimator of $\bar{x}^T \bar{z}$. In practice, choosing normalization is subsumed in selecting the regularization parameter for kernel methods.

We will give a sketch of the proof, complete details can be found in the appendix.

First, we have the following lemma regarding the estimator $k_{\mathcal{H}}^{\text{target}}(x_e, z_e)$.

**Lemma 6.** *Given two samples $x_e \sim N(\bar{x}, diag([\mathbf{0}_d, \sigma^2 I_{D-d}])), z_e \sim N(\bar{z}, diag([\mathbf{0}_d, \sigma^2 I_{D-d}])),$ let $k_{\mathcal{H}}(x_e, z_e) = x_e^T z_e$. We have:*

$$\mathbb{E}_{x_e, z_e}(k_{\mathcal{H}}^{\text{target}}(x_e, z_e)) = \bar{x}^T \bar{z} \ \text{ and } \ Var_{x_e, z_e}(k_{\mathcal{H}}^{\text{target}}(x_e, z_e)) = (D-d)\sigma^4.$$

Now we consider the Fredholm kernel with the help of unlabelled points from the distribution $p = N(\mathbf{0}, diag(\lambda^2 I_d, \sigma^2 I_{D-d}))$. Substituting $k_{\mathcal{H}}(u, v)$ by the linear kernel $u^T v$ in Eqn 9, we have:

$$k_F(x_e, z_e) = \int \int \frac{k(x_e, u)}{\int k(x_e, w)p(w)dw} \frac{k(z_e, v)}{\int k(z_e, w)p(w)dw} u^T v p(u)p(v)dudv$$

$$= \left( \frac{\int k(x_e, u)up(u)du}{\int k(x_e, w)p(w)dw} \right)^T \left( \int \frac{k(z_e, v)vp(v)dv}{\int k(z_e, w)p(w)dw} \right) \tag{11}$$

where recall $k(u, v) = \exp\left(-\frac{\|u-v\|^2}{2t}\right)$. Note $\frac{\int k(x_e,u)up(u)du}{\int k(x_e,w)p(w)dw}$ (resp. $\int \frac{k(z_e,v)vp(v)dv}{\int k(z_e,w)p(w)dw}$) is the weighted mean of the unlabeled data, with the weight function being the normalized Gaussian kernel centered at $x_e$ (resp. $z_e$). Hence by Eqn 11, $k_F(x_e, z_e)$ is the linear kernel between these two means (instead of the linear kernel for $x_e$ and $z_e$). Thus it is not too surprising that $k_F(x_e, z_e)$ should be more stable than the straightforward approximation $k_{\mathcal{H}}(x_e, z_e)$. Indeed, we have the following lemma (proof in appendix).

**Lemma 7.** *Given two samples $x_e \sim N(\bar{x}, diag([\mathbf{0}_d, \sigma^2 I_{D-d}])), z_e \sim N(\bar{z}, diag([\mathbf{0}_d, \sigma^2 I_{D-d}])),$ let $k_{\mathcal{H}}(x_e, z_e) = x_e^T z_e$ and $p = N(\mathbf{0}, diag(\lambda^2 I_d, \sigma^2 I_{D-d}))$. Let $k_F$ be as defined in Eqn 11. We have:*

$$\mathbb{E}_{x_e, z_e}\left( \left(\frac{t+\lambda^2}{\lambda^2}\right)^2 k_F(x_e, z_e) \right) = \bar{x}^T \bar{z}$$

*and*

$$Var_{x_e, z_e}\left( \left(\frac{t+\lambda^2}{\lambda^2}\right)^2 k_F(x_e, z_e) \right) = (D-d)\left( \frac{\sigma^2(t+\lambda^2)}{\lambda^2(t+\sigma^2)} \right)^4 \sigma^4$$

With Lemma 6 and 7, we can now compare the variances. Since $\frac{\sigma^2(t+\lambda^2)}{\lambda^2(t+\sigma^2)} < 1$ when $\lambda^2 > \sigma^2$, Theorem 5 follows.

Thus we can see the Fredholm kernel provides an approximation of the "true" linear kernel, but with smaller variance than the linear kernel on noisy data.

### 4.2.2 Gaussian Kernel

We now consider the case where the target kernel is the Gaussian kernel: $k_{\mathcal{H}}^{\text{target}}(u, v) = \exp\left(-\frac{\|u-v\|^2}{2r}\right)$. To approximate this kernel, we will set both $k$ and $k_{\mathcal{H}}$ to be Gaussian kernels. To simplify the presentation of results, we assume that $k$ and $k_{\mathcal{H}}$ have the same kernel width $t$. The resulting Fredholm kernel turns out to also be a Gaussian kernel, whose kernel width depends on the choice of $t$.

Our main result is the following. Again, similar to the case of linear kernel, the Fredholm estimator $k_F(x_e, z_e)$ and the vanilla one $k_{\mathcal{H}}^{\text{target}}(x_e, z_e)$ are both unbiased estimator for the target $k_{\mathcal{H}}^{\text{target}}(\bar{x}, \bar{z})$ upto a constant; but $k_F(x_e, z_e)$ has a smaller variance.

**Theorem 8.** *Suppose the probability distribution for the unlabeled data $p_X = N(\mathbf{0}, diag(\lambda^2 I_d, \sigma^2 I_{D-d}))$. Given the target kernel $k_{\mathcal{H}}^{\text{target}}(u, v) = \exp\left(-\frac{\|u-v\|^2}{2r}\right)$ with kernel width $r > 0$, we can choose t, given by the equation $\frac{t(t+\lambda^2)(t+3\lambda^2)}{\lambda^4} = r$, and two scaling constants $c_1, c_2$, such that*

$$\mathbb{E}_{x_e, z_e}(c_1^{-1} k_{\mathcal{H}}^{\text{target}}(x_e, z_e)) = \mathbb{E}_{x_e, z_e}(c_2^{-1} k_F(x_e, z_e)) = k_{\mathcal{H}}^{\text{target}}(\bar{x}, \bar{z}).$$

and when $\lambda^2 > \sigma^2$, we have $Var_{x_e, z_e}(c_1^{-1} k_{\mathcal{H}}^{target}(x_e, z_e)) > Var_{x_e, z_e}(c_2^{-1} k_F(x_e, z_e))$.

**Remark.** In practice, when applying kernel methods for real world applications, optimal kernel width $r$ is usually unknown and chosen by cross-validation or other methods. Similarly, for our Fredholm kernel, one can also use cross-validation to choose the optimal $t$ for $k_F$.

The proof of Theorem 8 is more complicated than in the linear case, and can be found in the appendix.

## 5 Experiments

In this section, we will demonstrate our Fredholm kernel empirically using both synthetic examples and data sets of text categorization and handwriting recognition. In section 5.1, we will use several examples to illustrate the effect of reducing variances using Fredholm kernel and how noise assumption is distinguished from the conventional assumptions in semi-supervised learning, such as cluster assumption and manifold assumption. In section 5.2, we show how classifiers based on Fredholm kernel perform on real world data sets like hand-written digits recognition and text categorization problems, compared with other semi-supervised algorithms.

First recall the Fredholm kernel we defined in previous section.

$$\hat{k}_F(x, z) = \frac{1}{(l+u)^2} \sum_{i,j=1}^{l+u} k(x, x_i) k_{\mathcal{H}}(x_i, x_j) k(z, x_j).$$

And using linear and Gaussian kernel for $k$ or $k_{\mathcal{H}}$, we can define three instances of the Fredholm kernel as follows.

(1) FredLin1: $k(x, z) = x^T z$ and $k_{\mathcal{H}}(x, z) = \exp\left(-\frac{\|x-z\|^2}{2r}\right)$.

(2) FredLin2: $k(x, z) = \exp\left(-\frac{\|x-z\|^2}{2r}\right)$ and $k_{\mathcal{H}}(x, z) = x^T z$.

(3) FredGauss: $k(x, z) = k_{\mathcal{H}}(x, z) = \exp\left(-\frac{\|x-z\|^2}{2r}\right)$.

For the kernels in (2) and (3) that use the Gaussian kernel as outside kernel $k$, intuitively we can also define their normalized version using the following definition,

$$\hat{k}_{F,n}(x, z) = \frac{1}{(l+u)^2} \sum_{i,j=1}^{l+u} \frac{k(x, x_i)}{\sum_n k(x, x_n)} k_{\mathcal{H}}(x_i, x_j) \frac{k(z, x_j)}{\sum_n k(z, x_n)}.$$

The resulting kernels are denoted by FredLin2(N) and FredGauss(N) respectively.

### 5.1 Synthetic Examples

Using specially designed toy examples, we could empirically verify the behavior of Fredholm kernel characterized by theoretical results in last section.

#### 5.1.1 Principal Component Regression

As we have pointed out in Theorem 4, Fredholm kernel and the associated Fredholm inner product space could stress the principal components with larger variances while suppressing the ones with smaller variances. Instead of hard cutting-off in many PCA-based methods, it provides a soft thresholding algorithm for feature selection. To demonstrate our methods, we consider the principal component regression problem [8], which assumes that the regressor $X$ and response $Y$ have the following relationship:

$$Y = \alpha X u_1,$$

wher $u_1$ is the first principal component. In this experiments, the data distribution is a Gaussian distribution $N(0, \text{diag}([10, 1, \ldots, 1]))$. Note that the axes themselves are the principal components. We will compare our method with linear regression using (1) all the dimensions; and (2) first $k$ principal components, while Fredholm kernel does not need to do any hard thresholding. Figure 2

shows the error of regression using Fredholm linear kernel or the projections to the first $k$ principal components. In the experiments, we uses 2000 unlabeled data for Fredholm kernel and PCA. The horizontal axis indicates different numbers of training points we used for training the regression. It can be observed that Fredholm kernel performs better than regression using the first $k$ principal components, unless the right number of principal components is chosen correctly, which is a non-trivial problem in practice.

Figure 2: The error of regression using Fredholm linear kernel or the projections to the first $k$ principal components.

### 5.1.2 Noise and cluster assumptions

Semi-supervised learning algorithms have shown better performance on various classification problems than the supervised learning algorithms. For example, [7] showed TSVM achieved the state of art performance on the problem of text categorization, and manifold regularization also showed good performance on various applications [1].

As we pointed out before, Fredholm kernel could deal with the noise assumption, which is distinct from the commonly used cluster assumption in many semi-supervised learning algorithms. To demonstrate our point, we use two toy examples that obviously violate the cluster assumption, shown in Figure 3. Each example is based on 1-dimensional manifold(s), and corrupted with additional Gaussian noise in $\mathbb{R}^{100}$. We assign the label to each point as we indicate in the figure by color. For each class, we will give a few labeled points, and large amount of unlabeled points from the marginal data distribution $p_X$. Since the data points are sampled around the underlying manifold, they served as two concrete examples of noise assumption, one for linear separable and the other for the non-linear separable case.

In our experiments, we compare Fredholm kernel based classifier with Regularizaed Least Square Classifier (RLSC), and two widely used semi-supervised methods, the transductive support vector machine (TSVM) and LapRLSC. Since the examples violate the cluster assumption, the two existing semi-supervised learning algorithms, TSVM and LapRLSC, should not gain much from the unlabeled data. For TSVM, we use the primal TSVM proposed in [3], since they claim primal TSVM usually performs better than the original algorithm in [7]; and we will use the implementation of LapRLSC given in [1]. For the linear separable case, linear classifiers are trained using these methods, while for the circular case, we will leverage Gaussian kernel to obtain a non-linear classifier. Similarly, we use the two linear Fredholm kernels introduced in Section 4.1 and 4.2.1, denoted by FredLlin1 and FredLin2, for the first toy example; and we use the double-Gaussian Fredholm kernel for the second circular toy example. Different numbers of labeled points are given for each class, together with another 2000 unlabeled points. To choose the optimal parameters for each method, we pick the parameters based on their performance on the validation set, while the final classification error is computed on the held-out testing data set. The classification error is presented in

(a)                                                    (b)

Figure 3: Two toy examples used to demonstrate the noise assumption.

Table 1 and 2, in which Fredholm kernels show clear improvement over other methods for synthetic examples in term of classification error.

| Number | Methods | | | | |
|---|---|---|---|---|---|
| of Labeled | RLSC | TSVM | LapRLSC | FredLin1 | FredLin2(N) |
| 8 | 10.0($\pm$ 3.9) | 5.2($\pm$ 2.2) | 10.0($\pm$ 3.5) | **3.7**($\pm$ 2.6) | 4.5($\pm$ 2.1) |
| 16 | 9.1($\pm$ 1.9) | 5.1($\pm$ 1.1) | 9.1($\pm$ 2.2) | **2.9**($\pm$ 2.0) | 3.6($\pm$ 1.9) |
| 32 | 5.8($\pm$ 3.2) | 4.5($\pm$ 0.8) | 6.0($\pm$ 3.2) | **2.3**($\pm$ 2.3) | 2.6($\pm$ 2.2) |

Table 1: The prediction error of the different classifiers on the linear toy example.

| Number | Methods | | | |
|---|---|---|---|---|
| of Labeled | RLSC | TSVM | LapRLSC | FredGauss(N) |
| 16 | 17.4($\pm$ 5.0) | 32.2($\pm$ 5.2) | 17.0($\pm$ 4.6) | **7.1**($\pm$ 2.4) |
| 32 | 16.5($\pm$ 7.1) | 29.9($\pm$ 9.3) | 18.0($\pm$ 6.8) | **6.0**($\pm$ 1.6) |
| 64 | 8.7($\pm$ 1.7) | 20.3($\pm$ 4.2) | 9.7($\pm$ 2.0) | **5.5**($\pm$ 0.7) |

Table 2: The prediction error of the different classifiers on the circular toy example.

## 5.2   Real-world Data Sets

Unlike toy examples, it is usually very difficult to verify whether certain assumption is satisfied or not in real problems. In this section, we will try to demonstrate the performance of Fredholm kernel on several real-world data sets and compare it with the baseline algorithms we used for toy examples. We organize the experiments by the kernel used for the classifiers. For example, in text categorization problem, linear kernel over the tfidf feature space usually gives great performance; and for handwriting digits recognition, Gaussian kernel usually performs better than using linear kernel. In the following experiments, we will apply several instances of Fredholm kernel to different data sets including text categorization and the handwritten digits recognition problem.

### 5.2.1   Linear Kernel

In this section, we will consider the problem of text categorization, which is a classic example for many semi-supervised learning problems. It labels each article or webpage by its topic. Recently, sentiment analysis has been another trending problem in text mining. It tries to categorize each short text, such as tweets or movie review, into positive or negative. And this problem is more subtle than the traditional text categorization, since sentiment is usually very tricky to detect and the text for this problem is usually shorter. In this experiment, we use the following 4 data sets from the literature:

(1) 20 news group: it has 11269 documents with 20 classes, and we select the first 10 categories for our experiment.

(2) Webkb: the original data set contains 7746 documents with 7 unbalanced classes, and we pick the two largest classes with 1511 and 1079 instances respectively.

(3) IMDB movie review: it has 1000 positive reviews and 1000 negative reviews of movie on IMDB.com.

(4) Twitter sentiment data set from Sem-Eval 2013: it contains 5173 tweets, with positive, neural and negative sentiment, and we combine neural and negative classes to make a relatively balanced binary classification problem.

For each data set, we extract tfidf from every document as the feature. Given the high dimensionality of tfidf feature in most cases, using linear kernel usually gives a great performance for text categorization problem.

For each data sets, we will use Fredholm kernels (1) and (2), which have a similar behavior with linear kernel, but perform much better. We will use the the purely supervised RLSC, and semi-superivsed Transductive SVM as baseline methods for comparison. Note that we use the implementation in [3] for TSVM, since they claim to achieve comparable performance while having a more simple algorithm using primal optimization.

To adapt the original data sets for the purpose of semi-supervised learning, we randomly pick-up a few points as labeled ones for each class, and use the rest of the data set as unlabeled points. And this splitting will be repeated for 10 times to estimate an average performance. Due to limited number of labeled points does not allow cross-validation, we pick the optimal parameter on testing data for all methods. The regularization parameter needs to be chosen for all methods, while we need to choose an extra kernel width for Fredholm kernel.

To measure the performance, we use the prediction error, the percentage of data gotten classified incorrectly. The experiments are given in Table 3. To further explore the influence of number of labeled points for each methods, we vary the amount of labeled points from 10 per class to 80 per class on Webkb data sets. And the performance for each methods is shown in Table 4.

| Data Set | Methods | | | | |
|---|---|---|---|---|---|
| | RLSC | TSVM | FredLin1 | FredLin2 | FredLin2(N) |
| Webkb | 16.9($\pm$ 1.4) | 12.7($\pm$ 0.8) | 13.0($\pm$ 1.3) | **12.0**($\pm$ 1.6) | **12.0**($\pm$ 1.6) |
| 20news | 22.2($\pm$ 1.0) | 21.0($\pm$ 0.9) | **20.5** ($\pm$ 0.7) | **20.5** ($\pm$0.7) | **20.5**($\pm$ 0.7) |
| IMDB | 30.0($\pm$ 2.0) | 20.2($\pm$ 2.6) | **19.9**($\pm$ 2.3) | 21.7($\pm$ 2.9) | 21.7($\pm$ 2.7) |
| Twitter | 38.7($\pm$ 1.1) | 37.6($\pm$ 1.4) | **37.4**($\pm$ 1.2) | **37.4**($\pm$ 1.2) | 37.5($\pm$ 1.2) |

Table 3: The error of various methods on the text data sets. 20 labeled data per class are given with rest of the data set as unlabeled points.

| Number of Labeled | Methods | | | | |
|---|---|---|---|---|---|
| | RLSC | TSVM | FredLin1 | FredLin2 | FredLin2(N) |
| 10 | 20.7($\pm$ 2.4) | **13.5**($\pm$ 0.5) | 14.8($\pm$ 2.4) | 14.6($\pm$ 2.4) | 14.6($\pm$ 2.3) |
| 20 | 16.9($\pm$ 1.4) | 12.7($\pm$ 0.8) | 13.0($\pm$ 1.3) | **12.0**($\pm$ 1.6) | **12.0**($\pm$ 1.6) |
| 80 | 10.9($\pm$ 1.4) | 9.7($\pm$ 1.0) | 8.1($\pm$ 1.0) | **7.9**($\pm$ 0.9) | **7.9**($\pm$ 0.9) |

Table 4: The prediction error on Webkb, with different number of labeled points, varying from 10 per class to 80 per class.

### 5.2.2 Gaussian Kernel

As we shown in Section 4.2.2, Fredholm kernel could also provide a more stable estimator for Gaussian kernel, when the Gaussian kernel is used for both $k$ and $k_{\mathcal{H}}$. To demonstrate this effect, we try to solve the problem of handwriting digits recognition. We choose this problem since it is non-linear separable and Gaussian kernel tends to give better performance than linear kernel empirically. The experiment uses subsets of two handwriting digits data sets MNIST and USPS: (1) the one from

| Data Set | Methods | | | |
|---|---|---|---|---|
| | KRLSC | LapRLSC | FredGauss | FredGauss(N) |
| USPST | 11.8($\pm$ 1.4) | **10.2** ($\pm$0.5) | 12.4($\pm$ 1.8) | 10.8($\pm$ 1.1) |
| MNIST | 14.3($\pm$ 1.2) | **8.6**($\pm$ 1.2) | 12.2($\pm$1.0) | 13.0($\pm$ 0.9) |

Table 5: The prediction error of nonlinear classifiers on the handwriting digits recognition data sets. 20 labeled data per class are given with rest of the data set as unlabeled points.

MNIST contains 10k digits in total with balanced examples for each class, and the one for USPS is the original testing set containing about 2k images. The pixel values are normalized to $[0, 1]$ as features.

For comparison, we also build classifiers using kernel RLSC and another semi-supervised algorithm, manifold regularization, which is known to perform very well on handwriting digits recognition when using Gaussian kernel. The results are presented in Table 5.

In Table 6, we show that as we add additional Gaussian noise to MNIST data, Fredholm kernels start to show significant improvement.

| Number of Labeled | Methods | | | |
|---|---|---|---|---|
| | KRLSC | LapRLSC | FredGauss | FredGauss(N) |
| 10 | 34.1($\pm$ 2.1) | 35.6 ($\pm$3.5) | **27.9**($\pm$ 1.6) | 29.0($\pm$ 1.5) |
| 20 | 27.2($\pm$ 1.1) | 27.3 ($\pm$1.8) | **21.9**($\pm$ 1.2) | 22.9($\pm$ 1.2) |
| 40 | 20.0($\pm$ 0.7) | 20.3 ($\pm$0.8) | **17.3**($\pm$ 0.5) | 18.4($\pm$ 0.4) |
| 80 | 15.6($\pm$ 0.4) | 15.6 ($\pm$0.5) | **14.8**($\pm$ 0.6) | 15.4($\pm$ 0.5) |

Table 6: The prediction error of nonlinear classifiers on MNIST corrupted with Gaussian noise with standard deviation 0.3 with different numbers of labeled points, from 10 to 80.

Note that we do not present the result for TSVM for this experiment, since an explicit feature map needs to constructed for the primal optimization. Such feature map is usually only an approximation, which might downgrade its performance.

### 5.3 Efficient Implementation Using Random Features

Even though kernel method has achieved significant success over the last decade, it usually suffers from the issue of scaling-up, due to the memory consumption quadratic to the size of the training data. It has inspired a line of research to solve this issue. For example, the random Fourier feature was proposed in [11]. This provides a way to efficiently approximate the several popular kernels, only requiring linear size of memory. Key idea of random Fourier features comes from the fact that every positive semi-definite kernel is the Fourier transform of a probability distribution by Bochner's theorem,

$$k(x - y) = \int_{\mathbb{R}^d} p(\omega)e^{i\omega'(x-y)}d\omega = \mathbb{E}_\omega(\xi_\omega(x)\xi_\omega(y)^*),$$

where $\xi_\omega(x) = e^{i\omega'x}$. For certain kinds of kernels, a set of samples, $(\omega_1, \ldots, \omega_D)$ could be drawn from $p(\omega)$, such that the expectation $\mathbb{E}_\omega$ could be estimated using a finite sum. Thus letting

$$z_\omega(x) = \sqrt{\frac{1}{D}}[\cos(w_1'x), \ldots, \cos(w_D'x), \sin(w_1'x), \ldots, \sin(w_D'x)],$$

we will have $\frac{1}{D}z_\omega(x)'z_\omega(y) \approx k(x, y)$. Taking the example of Gaussian kernel $k(x, y) = \exp(-\frac{\|x-y\|^2}{2t})$, the distribution $p(\omega) = \exp\left(-\frac{t\|\omega\|^2}{2d}\right)$.

Our Fredholm kernel could also leverage this technique to process the large scale data. Suppose the inside kernel $k_{\mathcal{H}}$ in the definition of Fredholm kernel is Gaussian kernel with kernel width $t$. Using the random Fourier feature, we will have a random feature map $z$ to approximate the kernel $k_{\mathcal{H}}$, such that $\frac{1}{D}z_\omega(u)'z_\omega(v) \approx k_{\mathcal{H}}(u, v)$. Plug this approximation into the definition of Fredholm kernel in

Eqn 5, we have

$$\hat{k}_F(x,z) \approx \frac{1}{(l+u)^2} \sum_{i,j=1}^{l+u} k(x,x_i) \left( \frac{1}{D} z_\omega(x_i)' z_\omega(x_j) \right) k(z,x_j)$$

$$= \frac{1}{D} \left( \frac{1}{(l+u)} \sum_{i=1}^{l+u} k(x,x_i) z_\omega(x_i) \right)^T \left( \frac{1}{(l+u)} \sum_{j=1}^{l+u} k(z,x_j) z_\omega(x_j) \right)$$

Thus, let

$$z_F(x) = \frac{1}{l+u} \sum_{i=1}^{l+u} k(x,x_i) z_\omega(x_i),$$

we will have $\hat{k}_F(x,z) \approx \frac{1}{D} z_F(x)' z_F(z)$. Using the approximation, we do not need to store the whole kernel matrix $K_{\mathcal{H}}$ of size $(l+u) \times (l+u)$. In this way, the memory usage will be reduced significantly. And it makes large scale learning using Fredholm kernel more feasible.

## Footnotes

[1]We will be using the square loss to simplify the exposition. Other loss functions can also be used in Eqn 3.

[2]We note that the term "Fredholm Kernel" has also been used before in a different context, see page 103, [6] and [16] in the studies of Fredholm operator. But our usage and the previous one represent different object.

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

## A  Proofs

### A.1  Lemma 1: The Representer's Theorem For Fredholm Learning

*Proof.* Define the empirical loss function for the learning problem,

$$L(f) = \frac{1}{l}\sum_{i=1}^{l}((\mathcal{K}_{\hat{p}_X}f)(x_i) - y_i)^2 + \lambda\|f\|_{\mathcal{H}}^2.$$

First let $\mathcal{H}_X$ be the sub-space of $\mathcal{H}$, spanned by kernel functions centered at the data points, $k_{\mathcal{H}}(x, x_i)$ for $i = 1, \ldots, l + u$. For the optimal solution of Eqn 3, we can have the orthogonal decomposition,

$$f^*(x) = f_X + f_{\perp},$$

where $f_X \in \mathcal{H}_X$ and $f_{\perp}$ is its orthogonal complement. By its definition, $f_{\perp}(x_i) = \langle f_{\perp}, k_{\mathcal{H}}(x, x_i)\rangle_{\mathcal{H}} = 0$ for $i = 1, \ldots, l + u$. Thus, the first term in the loss function $L(f)$ can be expanded as

$$\frac{1}{l}\sum_{i=1}^{l}((\mathcal{K}_{\hat{p}_X}f^*)(x_i) - y_i)^2 = \frac{1}{l}\sum_{i=1}^{l}\left(\frac{1}{l+u}k_{\mathcal{H}}(x_i, x_j)f^*(x_j) - y_i\right)^2$$

$$=\frac{1}{l}\sum_{i=1}^{l}\left(\frac{1}{l+u}\sum_{j=1}^{l+u}k_{\mathcal{H}}(x_i, x_j)(f_X(x_j) + f_{\perp}(x_j)) - y_i\right)^2 = \frac{1}{l}\sum_{i=1}^{l}\left(\frac{1}{l+u}\sum_{j=1}^{l+u}k_{\mathcal{H}}(x_i, x_j)f_X(x_j) - y_i\right)^2$$

$$=\frac{1}{l}\sum_{i=1}^{l}((\mathcal{K}_{\hat{p}_X}f_X)(x_i) - y_i)^2$$

So the orthogonal complement of $f_X$ does not matter at all for the empirical risk function.

For the regularization norm, we can use the pythagorean theorem in functional space.

$$\|f^*\|_{\mathcal{H}}^2 = \|f_X\|_{\mathcal{H}}^2 + \|f_{\perp}\|_{\mathcal{H}}^2,$$

thus, $\|f_X\|_{\mathcal{H}}^2 \le \|f^*\|_{\mathcal{H}}$. Combine the results above, we have

$$L(f_X) = \frac{1}{l}\sum_{i=1}^{l}((\mathcal{K}_{\hat{p}_X}f_X)(x_i) - y_i)^2 + \lambda\|f_X\|_{\mathcal{H}}^2$$

$$\le \frac{1}{l}\sum_{i=1}^{l}((\mathcal{K}_{\hat{p}_X}f_X)(x_i) - y_i)^2 + \lambda\left(\|f_X\|_{\mathcal{H}}^2 + \|f_{\perp}\|_{\mathcal{H}}^2\right) \le L(f^*).$$

By the definition of $f^*$, we have $f_X = f^*$. □

### A.2  Proposition 3: Positive Semi-definiteness Of Fredholm Kernel

*Proof.* For any given $z_1, \ldots, z_p$, we have the $p \times p$ Fredholm kernel matrix with

$$(K_F)_{mn} = \frac{1}{(l+u)^2}\sum_{i,j}k(z_m, x_i)k_{\mathcal{H}}(x_i, x_j)k(z_n, x_j).$$

Given a $p \times 1$ vector $\alpha$, we have

$$\boldsymbol{\alpha}^T K_F \boldsymbol{\alpha} = \sum_{m,n}\alpha_m\alpha_n\frac{1}{(l+u)^2}\sum_{i,j}k(z_m, x_i)k_{\mathcal{H}}(x_i, x_j)k(z_n, x_j)$$

$$=\frac{1}{(l+u)^2}\sum_{i,j}\left(\sum_m\alpha_m k(z_m, x_i)\right)\left(\sum_n\alpha_n k(z_n, x_j)\right)k_{\mathcal{H}}(x_i, x_j) \ge 0$$

due to the positive semi-definiteness of $k_{\mathcal{H}}$.

A similar argument can establish the same claim for the normalized version of Fredholm kernel in Eq 6. □

## A.3 Proof for Theorem 4

*Proof.* Recall that $\Sigma_F = \int \int uv^T k_{\mathcal{H}}(u,v)p(u)p(v)dudv$. Now substituting the distribution $p$ for unlabeled data and $k_{\mathcal{H}}$ as specified in the theorem, we have:

$$
\Sigma_F = \int \int uv^T k_{\mathcal{H}}(u,v)p(u)p(v)dudv
$$

$$
= \frac{1}{(2\pi)^D \prod_d \sigma_d^2} \int \int uv^T \prod_d \exp\left(-\frac{(u_d - v_d)^2}{2t}\right) \exp\left(-\frac{(u_d - \mu_d)^2}{2\sigma_d^2}\right) \exp\left(-\frac{(v_d - \mu_d)^2}{2\sigma_d^2}\right) dudv
$$

$\Sigma_F$ is a matrix, and we compute its entries separatedly. First, for the diagonal entries of $\Sigma_F$, we have for any $i \in [1, D]$,

$$
(\Sigma_F)_{ii} = \frac{1}{(2\pi)^d \prod_j \sigma_j^2} \int u_i v_i \prod_j \exp\left(-\frac{(u_j - v_j)^2}{2t}\right) \exp\left(-\frac{(u_j - \mu_j)^2}{2\sigma_j^2}\right) \exp\left(-\frac{(v_j - \mu_j)^2}{2\sigma_j^2}\right) dudv
$$

$$
= \frac{1}{(2\pi)^d \prod_j \sigma_j^2} \int u_i v_i \prod_j \exp\left(-\frac{(u_j - \frac{\sigma_j^2 v_j + t\mu_j}{t + \sigma_j^2})^2}{2\frac{t\sigma_j^2}{t+\sigma_j^2}}\right) \exp\left(-\frac{(v_j - \mu_j)^2}{2\frac{\sigma_j^2(t+\sigma_j^2)}{t+2\sigma_j^2}}\right) dudv
$$

$$
= \frac{1}{(2\pi)^{\frac{d}{2}}} \prod_j \frac{1}{\sigma_j^2} \left(\frac{t\sigma_j^2}{t + \sigma_j^2}\right)^{\frac{1}{2}} \int \left(\frac{\sigma_i^2 v_i^2 + t\mu_i v_i}{t + \sigma_i^2}\right) \prod_j \exp\left(-\frac{(v_j - \mu_j)^2}{2\frac{\sigma_j^2(t+\sigma_j^2)}{t+2\sigma_j^2}}\right) dv
$$

$$
= \prod_j \frac{1}{\sigma_j^2} \left(\frac{t\sigma_j^2}{t + \sigma_j^2} \frac{\sigma_j^2(t + \sigma_j^2)}{t + 2\sigma_j^2}\right)^{\frac{1}{2}} \left(\frac{\sigma_i^2\left(\mu_i^2 + \frac{\sigma_i^2(t+\sigma_i^2)}{t+2\sigma_i^2}\right) + t\mu_i^2}{t + \sigma_i^2}\right)
$$

$$
= \prod_j \sqrt{\frac{t}{t + 2\sigma_j^2}} \left(\mu_i^2 + \frac{\sigma_i^4}{t + 2\sigma_i^2}\right).
$$

For the off-diagonal entries, $(\Sigma_F)_{ij}$, $1 \le i \ne j \le D$, similar computation gives us the following

$$
(\Sigma_F)_{ij} = \frac{1}{(2\pi)^D \prod_d \sigma_d^2} \int \int u_i v_j \prod_d \exp\left(-\frac{(u_d - v_d)^2}{2t}\right) \exp\left(-\frac{(u_d - \mu_d)^2}{2\sigma_d^2}\right) \exp\left(-\frac{(v_d - \mu_d)^2}{2\sigma_d^2}\right) dudv
$$

$$
= \prod_d \sqrt{\frac{t}{t + 2\sigma_d^2}} \mu_i \mu_j.
$$

Put the above results together, we have the theorem. □

## A.4 Proof For Lemma 7

First, we need the following result and we include its proof for completeness.

**Lemma 9.** *Given a random variable $Z = X^T Y$, where $X, Y$ are two independent random vector, we have*

$$
\mathbb{E}(Z) = \mathbb{E}(X)^T \mathbb{E}(Y)
$$

*and*

$$
Var(Z) = \sum_{i=1}^{D} (\mathbb{E}(X_i)^2 Var(Y_i) + \mathbb{E}(Y_i)^2 Var(X_i) + Var(X_i)Var(Y_i))
$$

*Proof.* For expected value, we have

$$\mathbb{E}(Z) = \mathbb{E}(X^T Y) = \mathbb{E}\left(\sum_{i=1}^{D} X_i Y_i\right) = \sum_{i=1}^{D} \mathbb{E}(X_i Y_i) = \sum_{i=1}^{D} \mathbb{E}(X_i)\mathbb{E}(Y_i) = \mathbb{E}(X)^T \mathbb{E}(Y).$$

To compute variance, we first compute the second moment of $Z$,

$$\begin{aligned}
\mathbb{E}(Z^2) =& \mathbb{E}\left(\left(\sum_{i=1}^{D} X_i Y_i\right)^2\right) = \mathbb{E}\left(\sum_{i,j=1}^{D} X_i X_j Y_i Y_j\right) \\
=& \sum_{i\neq j} \mathbb{E}(X_i)\mathbb{E}(X_j)\mathbb{E}(Y_i)\mathbb{E}(Y_j) + \sum_{i=j} \mathbb{E}(X_i^2)\mathbb{E}(Y_i^2) \\
=& \sum_{i\neq j} \mathbb{E}(X_i)\mathbb{E}(X_j)\mathbb{E}(Y_i)\mathbb{E}(Y_j) + \sum_{i=1}^{D} (\mathbb{E}(X_i)^2 + Var(X_i))(\mathbb{E}(Y_i)^2 + Var(Y_i)) \\
=& (\mathbb{E}(X)^T \mathbb{E}(Y))^2 + \sum_{i=1}^{D} (\mathbb{E}(X_i)^2 Var(Y_i) + \mathbb{E}(Y_i)^2 Var(X_i) + Var(X_i)Var(Y_i))
\end{aligned}$$

Thus, the variance of $Z$ is

$$Var(Z) = \sum_{i=1}^{D} (\mathbb{E}(X_i)^2 Var(Y_i) + \mathbb{E}(Y_i)^2 Var(X_i) + Var(X_i)Var(Y_i))$$

$\square$

Now we can give the proof for **Lemma** 7.

*Proof.* By the assumption we have that the distribution $p$ for unlabelled points is a Gaussian distribution $N(0, \text{diag}([\lambda^2 I_d, \sigma^2 I_{D-d}]))$. Our goal is to compute the following $k_F(x_e, z_e)$.

$$\begin{aligned}
k_F(x_e, z_e) &= \int \int \frac{k(x_e, u)}{\int k(x_e, w)p(w)dw} \frac{k(z_e, v)}{\int k(z_e, w)p(w)dw} (u^T v) p(u)p(v)dudv \\
&= \left(\frac{\int k(x_e, u)up(u)du}{\int k(x_e, w)p(w)dw}\right)^T \left(\frac{\int k(z_e, v)vp(v)dv}{\int k(z_e, w)p(w)dw}\right) := (m_x)^T(m_z).
\end{aligned}$$

Note that we define $m_x, m_z$ to simplify the notations. And since $m_x, m_z$ are in the same form, we will only compute $m_x$, the formula for $m_z$ can be derived by the same computation. First, the denominator can be expended as

$$\int k(x_e, w)p(w)dw$$

$$= \frac{1}{(2\pi)^{D/2}(\lambda^2)^{d/2}(\sigma^2)^{(D-d)/2}} \int \prod_{i=1}^{D} \exp\left(-\frac{((x_e)_i - w_i)^2}{2t}\right) \prod_{i=1}^{d} \exp\left(-\frac{w_i^2}{2\lambda^2}\right) \prod_{i=d+1}^{D} \exp\left(-\frac{w_i^2}{2\sigma^2}\right) du$$

$$= \frac{1}{(2\pi)^{D/2}(\lambda^2)^{d/2}(\sigma^2)^{(D-d)/2}} \int \prod_{i=1}^{d} \exp\left(-\frac{(w_i - \frac{\lambda^2(x_e)_i}{t+\lambda^2})^2}{2\frac{t\lambda^2}{t+\lambda^2}}\right) \exp\left(-\frac{(x_e)_i^2}{2(t+\lambda^2)}\right)$$

$$\prod_{i=d+1}^{D} \exp\left(-\frac{(w_i - \frac{\sigma^2(x_e)_i}{t+\sigma^2})^2}{2\frac{t\sigma^2}{t+\sigma^2}}\right) \exp\left(-\frac{(x_e)_i^2}{2(t+\sigma^2)}\right) du$$

$$= \left(\frac{t}{t+\lambda^2}\right)^{d/2} \left(\frac{t}{t+\sigma^2}\right)^{(D-d)/2} \prod_{i=1}^{d} \exp\left(-\frac{(x_e)_i^2}{2(t+\lambda^2)}\right) \prod_{i=d+1}^{D} \exp\left(-\frac{(x_e)_i^2}{2(t+\sigma^2)}\right)$$

$$
\begin{aligned}
m_x =& \frac{\frac{1}{(2\pi)^{D/2}(\lambda^2)^{d/2}(\sigma^2)^{(D-d)/2}}}{\int k(x_e,w)p(w)dw} \int u k(x_e,u) \prod_{i=1}^{d} \exp\left(-\frac{u_i^2}{2\lambda^2}\right) \prod_{i=d+1}^{D} \exp\left(-\frac{u_i^2}{2\sigma^2}\right) du \\
=& \frac{\frac{1}{(2\pi)^{D/2}(\lambda^2)^{d/2}(\sigma^2)^{(D-d)/2}}}{\int k(x_e,w)p(w)dw} \int u \prod_{i=1}^{D} \exp\left(-\frac{((x_e)_i - u_i)^2}{2t}\right) \prod_{i=1}^{d} \exp\left(-\frac{u_i^2}{2\lambda^2}\right) \prod_{i=d+1}^{D} \exp\left(-\frac{u_i^2}{2\sigma^2}\right) du \\
=& \frac{\frac{1}{(2\pi)^{D/2}(\lambda^2)^{d/2}(\sigma^2)^{(D-d)/2}}}{\int k(x_e,w)p(w)dw} \int u \prod_{i=1}^{d} \exp\left(-\frac{(u_i - \frac{\lambda^2 (x_e)_i}{t+\lambda^2})^2}{2\frac{t\lambda^2}{t+\lambda^2}}\right) \exp\left(-\frac{(x_e)_i^2}{2(t+\lambda^2)}\right) \\
& \prod_{i=d+1}^{D} \exp\left(-\frac{(u_i - \frac{\sigma^2 (x_e)_i}{t+\sigma^2})^2}{2\frac{t\sigma^2}{t+\sigma^2}}\right) \exp\left(-\frac{(x_e)_i^2}{2(t+\sigma^2)}\right) du \\
=& [\frac{\lambda^2 (x_e)_1}{t+\lambda^2}, \ldots, \frac{\lambda^2 (x_e)_d}{t+\lambda^2}, \frac{\sigma^2 (x_e)_{d+1}}{t+\sigma^2}, \ldots, \frac{\sigma^2 (x_e)_D}{t+\sigma^2}] \\
=& \frac{\lambda^2}{t+\lambda^2}\bar{x} + \frac{\sigma^2}{t+\sigma^2}(x_e - \bar{x}).
\end{aligned}
$$

The last equation is because $x_e$ only has noises in the last $D-d$ coordinates, thus it has the same first $d$ coordinates with $\bar{x}$ up to a rescaling factor. Note that $x_e - \bar{x}$ is the noise term. If $t$ is significant large than the variance $\sigma^2$ for noise, then this noise term will be suppressed significantly. To apply Lemma 9, we need to compute the expected value and variance of $m_x$. It is easy to see:

$$
\mathbb{E}(m_x) = \frac{\lambda^2}{t+\lambda^2}\bar{x}.
$$

Since $x_e - \bar{x}$ accounts for the randomness of $m_x$, and since $x_e \sim N(\bar{x}, diag([\mathbf{0}_d, \sigma^2 I_{D-d}]))$, it follows that $Var((m_x)_i) = 0$ for $i \leq d$. For $d < i \leq D$, we have

$$
Var((m_x)_i) = \left(\frac{\sigma^2}{t+\sigma^2}\right)^2 \sigma^2.
$$

Applying Lemma 9, we have

$$
\mathbb{E}(m_x^T m_z) = \left(\frac{\lambda^2}{t+\lambda^2}\right)^2 \bar{x}^T \bar{z},
$$

and

$$
Var(m_x^T m_z) = (D-d)\left(\frac{\sigma^2}{t+\sigma^2}\right)^4 \sigma^4.
$$

(The derivation of the variance above uses the fact that $\bar{x}$ and $\bar{z}$ are located on the subspace of $\mathbb{R}^D$ spanned by the first $d$ axes.) Thus, by multiplying $k_F$ by the normalizing term $\left(\frac{t+\lambda^2}{\lambda^2}\right)^2$, we prove the theorem. $\qquad\square$

## A.5 Proof for Theorem 8

To prove this theorem, we first characterize the approximation $k_{\mathcal{H}}^{\text{target}}(x_e, z_e)$ using kernel $k_{\mathcal{H}}^{\text{target}}$, in term of its mean and variance, by the following lemma.

**Lemma 10.** *Given $\bar{x}, \bar{z}$, and two noise samples $x_e \sim N(\bar{x}, diag([\mathbf{0}_d, \sigma^2 I_{D-d}])), z_e \sim N(\bar{z}, diag([\mathbf{0}_d, \sigma^2 I_{D-d}]))$. Let $k_{\mathcal{H}}^{\text{target}}(x_e, z_e) = \exp\left(-\frac{\|x_e - z_e\|^2}{2r}\right)$ and $c_1 = \left(\frac{r}{r+2\sigma^2}\right)^{(D-d)/2}$, we have*

$$
\mathbb{E}_{x_e, z_e}(c_1^{-1} k_{\mathcal{H}}^{\text{target}}(x_e, z_e)) = \exp\left(-\frac{\|\bar{x} - \bar{z}\|^2}{2r}\right),
$$

*and*

$$
Var_{x_e, z_e}(c_1^{-1} k_{\mathcal{H}}^{\text{target}}(x_e, z_e)) = \left(\left(\frac{(r+2\sigma^2)^2}{r(r+4\sigma^2)}\right)^{(D-d)/2} - 1\right)\exp\left(-\frac{\|\bar{x} - \bar{z}\|^2}{r}\right)
$$

*Proof.* First of all, let us compute the expectation of $k_{\mathcal{H}}^{\text{target}}(x_e, z_e)$. Note that the first $d$ coordinates of $x_e, z_e$ are deterministic in our setting.

$$\mathbb{E}_{x_e, z_e}(k_{\mathcal{H}}^{\text{target}}(x_e, z_e))$$

$$= \int k_{\mathcal{H}}^{\text{target}}(x_e, z_e) p(x_e) p(z_e) dx_e dz_e$$

$$= \frac{1}{(2\pi\sigma^2)^{D-d}} \prod_{i=1}^{d} \exp\left(-\frac{((\bar{x})_i - (\bar{z})_i)^2}{2r}\right) \times$$

$$\int \prod_{i=d+1}^{D} \exp\left(-\frac{((x_e)_i - (z_e)_i)^2}{2r}\right) \exp\left(-\frac{(x_e)_i^2}{2\sigma^2}\right) \exp\left(-\frac{(z_e)_i^2}{2\sigma^2}\right) dx_e dz_e$$

$$= \frac{1}{(2\pi\sigma^2)^{D-d}} \exp\left(-\frac{\|\bar{x} - \bar{z}\|^2}{2r}\right) \times$$

$$\int \prod_{i=d+1}^{D} \exp\left(-\frac{\left((x_e)_i - \frac{\sigma^2(z_e)_i}{r+\sigma^2}\right)^2}{2\frac{r\sigma^2}{r+\sigma^2}}\right) \exp\left(-\frac{(z_e)_i^2}{2\frac{\sigma^2(r+\sigma^2)}{r+2\sigma^2}}\right) dx_e dz_e$$

$$= \frac{1}{(\sigma^2)^{D-d}} \left(\frac{r\sigma^2}{r+\sigma^2} \frac{\sigma^2(r+\sigma^2)}{r+2\sigma^2}\right)^{\frac{D-d}{2}} \exp\left(-\frac{\|\bar{x} - \bar{z}\|^2}{2r}\right)$$

$$= \left(\frac{r}{r+2\sigma^2}\right)^{\frac{D-d}{2}} \exp\left(-\frac{\|\bar{x} - \bar{z}\|^2}{2r}\right).$$

Thus, let $c_1 = \left(\frac{r}{r+2\sigma^2}\right)^{\frac{D-d}{2}}$, we have $\mathbb{E}_{x_e, z_e}(c_1^{-1} k_{\mathcal{H}}^{\text{target}}(x_e, z_e)) = \exp\left(-\frac{\|\bar{x} - \bar{z}\|^2}{2r}\right)$.

Similarly, we will get the second moment.

$$\mathbb{E}_{x_e, z_e}(k_{\mathcal{H}}^{\text{target}}(x_e, z_e)^2) = \left(\frac{r}{r+4\sigma^2}\right)^{\frac{D-d}{2}} \exp\left(-\frac{\|\bar{x} - \bar{z}\|^2}{r}\right).$$

Using $\text{Var}_{x_e, z_e}(c_1^{-1} k_{\mathcal{H}}^{\text{target}}(x_e, z_e)) = c_1^{-2} \left(\mathbb{E}_{x_e, z_e}(k_{\mathcal{H}}^{\text{target}}(x_e, z_e)^2) - \mathbb{E}_{x_e, z_e}(k_{\mathcal{H}}^{\text{target}}(x_e, z_e))^2\right)$, we have the result for variance. $\qquad\square$

Now consider the behavior of the Fredholm kernel. Under our specific setting, we know the distribution $p_X$, the integral in the definition of Fredholm kernel in Eq 9 could be computed explicitly. To keep our point clear, we omit the constant coefficient,

$$k_F(x_e, z_e) \propto \exp\left(-\frac{\|\bar{x} - \bar{z}\|^2}{2\frac{t(t+3\lambda^2)(t+\lambda^2)}{\lambda^4}}\right) \exp\left(-\frac{\|(x_e - \bar{x}) - (z_e - \bar{z})\|^2}{2\frac{t(t+3\sigma^2)(t+\sigma^2)}{\sigma^4}}\right)$$

$$= \exp\left(-\frac{\|x_0 - z_0\|^2}{2\frac{t(t+3\lambda^2)(t+\lambda^2)}{\lambda^4}}\right)$$

where $x_0 = \bar{x} + \eta(x_e - \bar{x})$, $z_0 = \bar{z} + \eta(z_e - \bar{z})$, and $\eta = \frac{\sigma^4(st+s\lambda^2+2t\lambda^2)(t+\lambda^2)}{\lambda^4(st+s\sigma^2+2t\sigma^2)(t+\sigma^2)}$. Since $\sigma^2$ is the variance for noise, $\sigma^2 < \lambda^2$, and thus $\eta < 1$. It can be observed that the resulting Fredholm kernel is still a Gaussian kernel. By selecting $t$ properly, the kernel width could match the original kernel, while the center of new kernel, $x_0, z_0$, becomes closer to $x, z$ than the original center $x_e, z_e$. Intuitively, this Fredholm kernel gives a more stable elstimator for $k_{\mathcal{H}}^{\text{target}}$.

To formulate this idea strictly, we have the following lemma.

**Lemma 11.** *Given $\bar{x}, \bar{z}$, and two noise sample $x_e \sim N(\bar{x}, diag([\mathbf{0}_d, \sigma^2 I_{D-d}]))$, $z_e \sim N(\bar{z}, diag([\mathbf{0}_d, \sigma^2 I_{D-d}]))$. Suppose distribution of unlabeled data is $N(0, diag([\lambda^2 I_d, \sigma^2 I_{D-d}]))$. Letting $c_2 = \left(\frac{t(t+\sigma^2)^2}{t^3+4t^2\sigma^2+3t\sigma^4+2\sigma^6}\right)^{(D-d)/2} \left(\frac{t(t+\lambda^2)}{t(t+3\lambda^2)}\right)^{d/2}$, we have*

$$\mathbb{E}_{x_e, z_e}(c_2^{-1} k_F(x_e, z_e)) = \exp\left(-\frac{\|\bar{x} - \bar{z}\|^2}{2\frac{t(t+\lambda^2)(t+3\lambda^2)}{\lambda^4}}\right),$$

*and*

$$Var_{x_e,z_e}(c_2^{-1}k_F(x_e,z_e)) =$$
$$\left(\left(\frac{(t^3 + 4t^2\sigma^2 + 3t\sigma^4 + 2\sigma^6)^2}{t(t+\sigma^2)(t+3\sigma^2)(t^3 + 4t^2\sigma^2 + 3t\sigma^4 + 4\sigma^6)}\right)^{(D-d)/2} - 1\right) \exp\left(-\frac{\|\bar{x}-\bar{z}\|^2}{\frac{(t+\lambda^2)(t^2+3t\lambda^2)}{\lambda^4}}\right)$$

We can see that the difference between Fredholm kernel and the original kernel $k_{\mathcal{H}}^{\text{target}}$ is the kernel width. Thus we can choose $t$ and $s$ properly in Fredholm kernel such that the kernel width matches the one in $k_{\mathcal{H}}^{\text{target}}$ before comparing the variances.

Now we can give the proof for Theorem 8.

*Proof.* First, by setting $r = \frac{(t+\lambda^2)(st+s\lambda^2+2t\lambda^2)}{\lambda^4}$, we make the two approximations have the same expected value. Thus, it suffices to compare the variances of the adjusted approximations. With the $r$ plugged into the variance in Proposition 10, it suffices to show that

$$\frac{(\frac{(t+\lambda^2)(st+s\lambda^2+2t\lambda^2)}{\lambda^4} + 2\sigma^2)^2}{\frac{(t+\lambda^2)(st+s\lambda^2+2t\lambda^2)}{\lambda^4}(\frac{(t+\lambda^2)(st+s\lambda^2+2t\lambda^2)}{\lambda^4} + 4\sigma^2)} >$$
$$\frac{(st^2 + 2st\sigma^2 + 2t^2\sigma^2 + s\sigma^4 + 2t\sigma^4 + 2\sigma^6)^2}{(t+\sigma^2)(st + s\sigma^2 + 2t\sigma^2)(st^2 + 2st\sigma^2 + 2t^2\sigma^2 + s\sigma^4 + 2t\sigma^4 + 4\sigma^6)}$$
$$= \frac{(\frac{(t+\sigma^2)(st+s\sigma^2+2t\sigma^2)}{\sigma^4} + 2\sigma^2)^2}{\frac{(t+\sigma^2)(st+s\sigma^2+2t\sigma^2)}{\sigma^4}(\frac{(t+\sigma^2)(st+s\sigma^2+2t\sigma^2)}{\sigma^4} + 4\sigma^2)}$$

Since we have $\frac{(t+\sigma^2)(st+s\sigma^2+2t\sigma^2)}{\sigma^4} > \frac{(t+\lambda^2)(st+s\lambda^2+2t\lambda^2)}{\lambda^4}$ and the function $\frac{r+2\sigma^2}{r(r+4\sigma^2)}$ is decreasing w.r.t. $r$, we have the inequality. $\qquad\square$

### A.5.1 Proof For Lemma 11

Here, we will prove the general case that uses different kernel widths for $k$ and $k_{\mathcal{H}}$. Then one can simply set them to be the same to get **Lemma 11**.

Here's the new Lemma we will prove.

**Lemma 12.** *Given $\bar{x}, \bar{z}$, and two noise sample $x_e \sim N(\bar{x}, diag([\mathbf{0}_d, \sigma^2 I_{D-d}])), z_e \sim N(\bar{z}, diag([\mathbf{0}_d, \sigma^2 I_{D-d}]))$. Suppose distribution of unlabeled data is $N(0, diag([\lambda^2 I_d, \sigma^2 I_{D-d}]))$. Thus, we have*

$$\mathbb{E}_{x_e,z_e}(k_F(x_e,z_e)) = \left(\frac{s(t+\sigma^2)^2}{st^2 + 2st\sigma^2 + 2t^2\sigma^2 + s\sigma^4 + 2t\sigma^4 + 2\sigma^6}\right)^{(D-d)/2} \left(\frac{s(t+\lambda^2)}{st + s\lambda^2 + 2t\lambda^2}\right)^{d/2}$$
$$\exp\left(-\frac{\|\bar{x}-\bar{z}\|^2}{2\frac{(t+\lambda^2)(st+s\lambda^2+2t\lambda^2)}{\lambda^4}}\right).$$

*Let $c_2 = \left(\frac{s(t+\sigma^2)^2}{st^2+2st\sigma^2+2t^2\sigma^2+s\sigma^4+2t\sigma^4+2\sigma^6}\right)^{(D-d)/2} \left(\frac{s(t+\lambda^2)}{st+s\lambda^2+2t\lambda^2}\right)^{d/2}$. We have*

$$Var_{x_e,z_e}(c_2^{-1}k_F(x_e,z_e)) =$$
$$\left(\left(\frac{(st^2 + 2st\sigma^2 + 2t^2\sigma^2 + s\sigma^4 + 2t\sigma^4 + 2\sigma^6)^2}{(t+\sigma^2)(st+s\sigma^2+2t\sigma^2)(st^2+2st\sigma^2+2t^2\sigma^2+s\sigma^4+2t\sigma^4+4\sigma^6)}\right)^{(D-d)/2} - 1\right)$$
$$\exp\left(-\frac{\|\bar{x}-\bar{z}\|^2}{\frac{(t+\lambda^2)(st+s\lambda^2+2t\lambda^2)}{\lambda^4}}\right)$$

*Proof.* Again, since we know the exact distribution of the unlabeled data, we can compute the close formula of $k_F(x_e, z_e)$.

$$k_F(x_e, z_e) = \int\int \frac{k(x_e, u)}{\int k(x_e, w)p(w)dw} \frac{k(z_e, v)}{\int k(z_e, w)p(w)dw} k_{\mathcal{H}}(u, v)p(u)p(v)dudv$$

$$= \left(\frac{s(t+\lambda^2)}{st+s\lambda^2+2t\lambda^2}\right)^{d/2} \left(\frac{s(t+\sigma^2)}{st+s\sigma^2+2t\sigma^2}\right)^{(D-d)/2} \times$$

$$\exp\left(-\frac{\|\bar{x}-\bar{z}\|^2}{2\frac{(st+s\lambda^2+2t\lambda^2)(t+\lambda^2)}{\lambda^4}}\right) \exp\left(-\frac{\|(x_e-\bar{x})-(z_e-\bar{z})\|^2}{2\frac{(st+s\sigma^2+2t\sigma^2)(t+\sigma^2)}{\sigma^4}}\right)$$

Based on this computation, we need to compute expected value and variance of $k_F$. Note that the randomness of $k_F(x_e, z_e)$ comes from the term $x_e - \bar{x}$ and $z_e - \bar{z}$, we take out the random variable from the above formula, and denote it

$$Z = \exp\left(-\frac{\|(x_e-\bar{x})-(z_e-\bar{z})\|^2}{2\frac{(st+s\sigma^2+2t\sigma^2)(t+\sigma^2)}{\sigma^4}}\right).$$

Recall that the distributions for $x_e$ and $z_e$ are $N(\bar{x}, \text{diag}([\mathbf{0}, \sigma^2 I_{D-d}]))$ and $N(\bar{z}, \text{diag}([\mathbf{0}, \sigma^2 I_{D-d}]))$ respectively. For expected value, we have

$$\mathbb{E}_{x_e, z_e}(Z) = \int \exp\left(-\frac{\|(x_e-\bar{x})-(z_e-\bar{z})\|^2}{2\frac{(st+s\sigma^2+2t\sigma^2)(t+\sigma^2)}{\sigma^4}}\right) p(x_e)p(z_e)dx_edz_e$$

$$= \left(\frac{(t+\sigma^2)(st+s\sigma^2+2t\sigma^2)}{st^2+2st\sigma^2+2t^2\sigma^2+s\sigma^4+2t\sigma^4+2\sigma^6}\right)^{(D-d)/2}$$

And for the second moment, we have

$$\mathbb{E}_{x_e, z_e}(Z^2) = \int \exp\left(-\frac{\|(x_e-\bar{x})-(z_e-\bar{z})\|^2}{\frac{(st+s\sigma^2+2t\sigma^2)(t+\sigma^2)}{\sigma^4}}\right) p(x_e)p(z_e)dx_edz_e$$

$$= \left(\frac{(t+\sigma^2)(st+s\sigma^2+2t\sigma^2)}{st^2+2st\sigma^2+2t^2\sigma^2+s\sigma^4+2t\sigma^4+4\sigma^6}\right)^{(D-d)/2}$$

Thus,

$$Var(Z) = \mathbb{E}(Z^2) - \mathbb{E}(Z)^2 =$$

$$\left(\frac{(t+\sigma^2)(st+s\sigma^2+2t\sigma^2)}{st^2+2st\sigma^2+2t^2\sigma^2+s\sigma^4+2t\sigma^4+4\sigma^6}\right)^{(D-d)/2} - \left(\frac{(t+\sigma^2)(st+s\sigma^2+2t\sigma^2)}{st^2+2st\sigma^2+2t^2\sigma^2+s\sigma^4+2t\sigma^4+2\sigma^6}\right)^{(D-d)}$$

Now we multiply $Z$ by the constant term, we have

$$\mathbb{E}_{x_e, z_e}(k_F(x_e, z_e)) = \left(\frac{s(t+\sigma^2)^2}{st^2+2st\sigma^2+2t^2\sigma^2+s\sigma^4+2t\sigma^4+2\sigma^6}\right)^{(D-d)/2} \left(\frac{s(t+\lambda^2)}{st+s\lambda^2+2t\lambda^2}\right)^{d/2}$$

$$\exp\left(-\frac{\|\bar{x}-\bar{z}\|^2}{2\frac{(t+\lambda^2)(st+s\lambda^2+2t\lambda^2)}{\lambda^4}}\right).$$

And let $c_2 = \left(\frac{s(t+\sigma^2)^2}{st^2+2st\sigma^2+2t^2\sigma^2+s\sigma^4+2t\sigma^4+2\sigma^6}\right)^{(d-l)/2} \left(\frac{s(t+\lambda^2)}{st+s\lambda^2+2t\lambda^2}\right)^{l/2}$, we will have the results for the variance by scaling the variance of $Z$ by the constant term. $\qquad\square$