[Reviews · NeurIPS 2014]

Submitted by Assigned_Reviewer_5

This paper describes a framework particularly useful for semi-supervised learning based on Fredholm kernels. The classical supervised learning optimization problem solved in kernel-based methods is extended to incorporate unlabeled information leading to discretized version of the Fredholm integral equation.

Quality

The paper has high technical quality with well-supported claims by theoretical analysis and convincing experimental results. The proposed formulation leads to a new data-dependent kernel that incorporates unlabeled information. The classifying function differs from the classical represent theorem solution but it is still elegant and easy to compute.

I have few comments: One is related to Equation 3 where the solution of the proposed optimization problem is described. Since the Authors mentioned that the associated kernel need not be PSD, does Equation 3 still hold in this case ? A proof on how Equation 3 was obtained could have been nice to see too. Another comment concerns the experimental section where the Authors chose an "optimal" parameter for all methods in the benchmark. How were those parameters determined ? What is meant with "optimal" ?

Clarity

The paper is clearly written and well-organized in most part. It is easy to follow and the main ideas are explain adequately. There are some typos though for example:

* Line 82: "... will a Euclidean ..." -> ... will be a Euclidean ..."
* Line 83: "or" missing.
* Section 4 title: "Fredhom" -> Fredholm
* Line 247: "porportional" -> proportional
* Line 283: "That is, or..." -> That is, for...
* Line 352: Sentence needs to be rewritten.

Originality

The proposed approach is related to a recent NIPS publication ([9]) but it is novel in essence. It is clear how the work differs from this publication and others.

Significance

The results are important and significant. There was an significant effort to test the proposed method on several datasets in different application domains.
Summary: Well-written paper with good theoretical basis and convincing experimental section.

Submitted by Assigned_Reviewer_21

The authors present an interesting new method for supervised and semi-supervised learning using Fredholm kernels. Many aspects of the method are discussed in detail. Overall, I find the paper very interesting.

Nevertheless, I still have a few suggestions and questions:

- section 2 (from eq 1 to eq on line 99):

Also in the Nystrom method one considers an approximation to an integral equation
(in that case kernel PCA). I think it would be good if the authors could mention this and also explain in what sense the approach is similar or different.

- section 2 eq 2:

In the area of support vector machines and kernel methods also approaches with operator equations have been presented in literature. It would be good to mention this work and explain in which sense eq 2 is a special case of it (or explain how it differs).

- section 2 line 128 representer theorem:

Eq 2 is less common in learning theory. The authors assume that a representer theorem exists. They should prove a representer theorem here or otherwise give a precise reference and explain how eq 2 is a special case of it.

- section 4 eq 8:

The interpretation as soft-thresholding PCA is nice and interesting.
How does it differ from other methods like Fisher kernels and probability product kernels?

- typos:

line 165 Not that
line 221 achived
line 341 chossing
line 672 Table ??

Summary: New approach to supervised and unsupervised learning using Fredholm kernels, which are data-dependent.

Submitted by Assigned_Reviewer_28

The authors proposed a data dependent kernel called Fredholm kernels. The novelty of the kernel is to use unlabeled dataset in natural way. Authors showed that several theoretical property of the kernel and experimentally verified that the kernel compares favorably with existing state-of-the-art method.

The paper is well written and organized.

Detailed comments:
1. Is there any systematic approach to select kernel width for freadholm kernel? For example, under some assumption, can you select a kernel width by cross validation?
2. It would be nice to add some toy experiments to show the case the proposed method might not work. I just want to know in which setting the proposed method is most effective.
Summary: The paper is interesting and is very clearly written. I vote for an acceptance.
Author Feedback
Author rebuttal: We thank all reviewers for the thoughtful comments. We will incorporate them into the future revisions of this paper.

Reviewer 21:

The integral operator does play an important role in many machine learning algorithms. Nystrom method can be used to approximate the kernel and in fact can be combined with our approach to improve computational efficiency. We will add more discussion in the revision.

For Eq 2 and 3, we will provide more discussion of the related work. As far as the Representer Theorem is concerned, the proof uses the standard techniques. We will add it for completeness in the appendix of the revision.

Thanks for the interesting pointer to Fisher and probability-product kernels. We will look into that.

Reviewer 28:

Indeed the selection of kernel width is important for our method and it is not a trivial problem. However, the same difficulty applies to all kernel methods. The good news is that several of our methods do not increase the number of parameters compared to the standard kernel methods, while potentially achieving better performance using unlabeled data. If the size of the training data is sufficient, then cross-validation can be used.

As for the second question – our method has the most advantage over graph-based methods (e.g., LapSVM), i.e, is most effective, when the data satisfies the noise assumption but not the cluster assumption.

Reviewer 5:

Note that we need two kernels (K and K_H) to define our Fredholm kernel, and only one of them (K_H) has to be PSD for the resulting kernel to be PSD as well (see Proposition 1). Similarly Eq. 3 holds if K_H is PSD. The proof uses standard techniques. We will add it in the appendix of the revision.

Parameter selection is a hard problem, especially when the number of training data is small. To have a fair comparison between the algorithms, we choose the parameters which give best performance on a separate validation set for each algorithm. We will clarify that in the paper.